# Kv7/KCNQ potassium channels in cortical hyperexcitability and juvenile seizure-related death in Ank2-mutant mice

Hyoseon Oh [1,2], Suho Lee [2], Yusang Oh[1,3], Seongbin Kim [1], Young Seo Kim[4], Yeji Yang[1,5], Woochul Choi [3], Ye-Eun Yoo[2], Heejin Cho[2], Seungjoon Lee [2], Esther Yang[6], Wuhyun Koh [7], Woojin Won[7], Ryunhee Kim[2], C. Justin Lee [7], Hyun Kim [6], Hyojin Kang[8], Jin Young Kim [5], Taeyun Ku [4], Se-Bum Paik [3] & Eunjoon Kim [1,2] ✉

Autism spectrum disorders (ASD) represent neurodevelopmental disorders characterized by social deficits, repetitive behaviors, and various comorbidities, including epilepsy. *ANK2*, which encodes a neuronal scaffolding protein, is frequently mutated in ASD, but its in vivo functions and disease-related mechanisms are largely unknown. Here, we report that mice with *Ank2* knockout restricted to cortical and hippocampal excitatory neurons (Ank2-cKO mice) show ASD-related behavioral abnormalities and juvenile seizure-related death. Ank2-cKO cortical neurons show abnormally increased excitability and firing rate. These changes accompanied decreases in the total level and function of the Kv7.2/KCNQ2 and Kv7.3/KCNQ3 potassium channels and the density of these channels in the enlengthened axon initial segment. Importantly, the Kv7 agonist, retigabine, rescued neuronal excitability, juvenile seizure-related death, and hyperactivity in Ank2-cKO mice. These results suggest that Ank2 regulates neuronal excitability by regulating the length of and Kv7 density in the AIS and that Kv7 channelopathy is involved in Ank2-related brain dysfunctions.

Autism spectrum disorders (ASD) are characterized by social deficits, repetitive behaviors, and various comorbidities, including intellectual disability, anxiety, hyperactivity, and epilepsy. Previous studies identified various ASD-risk genes[1–3] and multiple underlying mechanisms, including synaptic deficits[4–7]. However, the converging mechanisms are still unclear, even for the most frequently mutated ASD-risk genes.

Ank2, also known as Ankyrin-B, is a scaffolding/adapter protein that anchors various surface membrane proteins, such as ion channels and cell adhesion molecules, to cytoskeletal proteins (e.g., spectrin and F-actin) at specialized subcellular compartments[8]. Ank2 has long been associated with cardiac dysfunctions[9] and, more recently, emerged as one of the most frequently mutated ASD-risk genes[2,3].

Previous studies on *Ank2*-deficient mice revealed that Ank2 regulates aspects of neuronal development, such as neurite initiation, axonal development, and neuronal connectivity through mechanisms that include the interaction of Ank2 with the cell adhesion molecule, L1CAM[10–13]. Ank2 was also shown to regulate the length of the axon initial segment (AIS), which critically regulates neuronal excitability[14,15],

[1]Department of Biological Sciences, Korea Advanced Institute for Science and Technology (KAIST), Daejeon 34141, Korea. [2]Center for Synaptic Brain Dysfunctions, Institute for Basic Science (IBS), Daejeon 34141, Korea. [3]Department of Bio and Brain Engineering, KAIST, Daejeon 34141, Korea. [4]Graduate School of Medical Science and Engineering, KAIST, Daejeon 34141, Korea. [5]Research Center for Bioconvergence Analysis, Korea Basic Science Institute, 162 Yeongudanjiro, Ochang, Cheongju, Chungbuk 28119, Korea. [6]Department of Anatomy and Brain Korea 21 Graduate Program, Biomedical Science, College of Medicine, Korea University, Seoul 02841, Korea. [7]Center for Cognition and Sociality, IBS, Daejeon 34126, Korea. [8]Division of National Supercomputing, Korea Institute of Science and Technology Information, Daejeon 34141, Korea. ✉e-mail: kime@kaist.ac.kr

by surrounding and thus defining the proximal and distal boundaries of the AIS in the axon[16]. However, it remains unclear how the length of the AIS, where ion channels are highly concentrated, could be functionally linked to the regulation of neuronal excitability and whether and how Ank2 is associated with disease-related pathophysiological conditions.

We found here that global *Ank2* heterozygosity fails to induce behavioral deficits but that *Ank2* homozygous deletion restricted to cortical and hippocampal excitatory neurons (Ank2-cKO neurons) leads to ASD-related behavioral deficits and juvenile seizure-related death. Ank2 cKO increased cortical neuronal excitability and firing through decreases in the level and function of Kv7.2/3 (or KCNQ2/3) potassium channels. Ank2 cKO also increased AIS length and decreased Kv7 density in the AIS. The FDA-approved Kv7 agonist, retigabine, rescued neuronal excitability and juvenile seizure-related death in Ank2-cKO mice. Therefore, Ank2 regulates neuronal excitability by controlling AIS length and density of Kv7, and Kv7 channelopathy is involved in Ank2-related brain dysfunctions.

## Results

### Behavioral deficits and juvenile seizure-related death in Ank2-cKO mice

To explore the mechanisms underlying *Ank2* deficiency-induced ASD, we generated mice carrying a floxed exon 4 of the *Ank2* gene (*Ank2^{fl/fl}* mice), which encodes the N-terminal region of the protein shared by the short and long Ank2 protein variants (220 and 440 kDa, respectively) (Fig. 1a; Supplementary Fig. 1a, b). We chose to delete both the short and long variants, which resembles some human *ANK2* mutations, to induce stronger abnormalities in mice. In addition, the floxed *Ank2* gene was generated to try both global and conditional gene knockouts (KOs). In situ hybridization detected the *Ank2* mRNA in various brain regions, including the cortex, hippocampus, and cerebellum, at embryonic and postnatal stages, and also in both excitatory and inhibitory neurons (Supplementary Fig. 1c, d). The majority of glutamatergic and GABAergic neurons in the cortex and hippocampus expressed *Ank2* mRNAs (Supplementary Fig. 1d–f). In addition, greater proportions of Ank2-positive neurons were glutamatergic than GABAergic.

Analyses of global *Ank2* heterozygous KO mice (*Ank2^{+/−}* mice; male), derived from *Ank2^{fl/fl}* mice (see Methods for details), using a battery of behavioral tests revealed that their behaviors were largely normal, compared with wild-type (WT) mice, in domains including locomotion, anxiety-like behavior, social interaction, repetitive behavior, learning and memory (spatial and fear), motor coordination, and sensory-motor behavior (Supplementary Fig. 2). Female global *Ank2^{+/−}* mice also showed largely normal behaviors in locomotor, anxiety-like, social, repetitive, and sensory-motor domains (Supplementary Fig. 3). However, there were two distinct differences: decreased open-field center time (anxiety-like behavior) in male but not female mutants and suppressed self-grooming in female but not male mutants.

Similar to previous reports[11,17], we found that global homozygous *Ank2* exon 4 deletion (*Ank2^{−/−}*) led to lethality in mice. We thus tested if homozygous *Ank2* deletion restricted to cortical and hippocampal excitatory neurons in *Emx1-Cre;Ank2^{fl/fl}* mice (termed Ank2-cKO hereafter), which decreased the Ank2 protein level to ~9–15% of the control (*Ank2^{fl/fl}*) level (Fig. 1b), would yield detectable phenotypes. The residual Ank2 proteins may represent those that are expressed in *Emx1*-negative cell types such as GABAergic inhibitory neurons since *Emx1-Cre* drives gene deletion in excitatory neurons and glial cells but not in inhibitory neurons[18].

Behaviorally, Ank2-cKO mice showed hyperactivity and anxiety-like behavior in the open-field test but anxiolytic-like behaviors in light-dark and elevated plus-maze tests (Fig. 1c–e), likely because of the distinct nature of the anxiogenic stimuli (open space, height, and light). These changes did not correlate with the hyperactivity of the

mutant mice (Supplementary Fig. 4a–e), suggesting that hyperactivity did not confound the results. Ank2-cKO mice showed largely normal social interaction but impaired social novelty recognition in the three-chamber test (Fig. 1f). These mice also displayed abnormally increased direct social interaction and decreased repetitive behaviors (self-grooming and digging) (Fig. 1g, h). The decreased repetitive behaviors did not correlate with the hyperactivity (Supplementary Fig. 4f, g).

Importantly, Ank2-cKO mice showed spontaneous epileptiform discharges in the frontal and parietal lobes, and juvenile seizure-related complete death occurring from postnatal day (P) 20 to P50 (Fig. 2a–c). In addition, when seizures were induced by pentylenetetrazole (PTZ), a GABA_A receptor antagonist, Ank2-cKO mice showed stronger levels of seizures (Fig. 2d–h), as shown by the final seizure stage (1–4) reached, latency to seizure stage 2, the number of animals reached stage 2, and the frequency and duration of stage 2.

These results collectively suggest that *Ank2* deletion restricted to cortical and hippocampal excitatory neurons leads to ASD-related behavioral abnormalities and increased spontaneous and induced seizures and juvenile death.

### Increased excitability and firing in Ank2-cKO cortical neurons

To explore the mechanisms underlying the behavioral and seizure and juvenile death phenotypes in Ank2-cKO mice, we characterized neuronal excitability and synaptic transmission in the somatosensory cortex (SSC), which is a brain region that has been frequently implicated in ASD[19,20] and epilepsy[21].

To this end, we first measured neuronal firing in brain slices containing the Ank2-cKO SSC using a multi-electrode array (MEA) (Fig. 3a) and found that the mean firing rates were increased in layers 2/3 and layer 5 (Fig. 3b). The frequency of bursts (inter-spike-interval <10 ms), a seizure-related measure, was also increased in layers 2/3 but not in layer 5 of Ank2-cKO brains (Fig. 3c); this increase mainly involved enhancement of the burst-event number rather than the number of action potentials (APs) per burst or the percentage of spikes in bursts (Fig. 3d, e). Local field potentials (LFPs), representing slow components of the MEA recordings, were also moderately changed in layers 2/3 but not in layer 5, as shown by the increased duration but unaltered frequency and amplitude of LFPs (Fig. 3f–i).

Patch-clamp recording indicated that there was increased intrinsic excitability in SSC layer 2/3 pyramidal neurons, as shown by the input resistance and current-firing curve, but no change in the resting membrane potential (Fig. 4a–d). The medium afterhyperpolarization (mAHP) after sustained APs was decreased (Fig. 4e, f), whereas AP threshold and AP shape-related parameters were unaltered (Fig. 4g–l).

Ank2-cKO layer 2/3 pyramidal neurons showed normal spontaneous excitatory synaptic transmission in the absence or presence of network activity (mEPSCs [miniature excitatory postsynaptic currents] and sEPSCs [spontaneous EPSCs]) (Supplementary Fig. 5a, c). mIPSCs (miniature inhibitory postsynaptic currents) were moderately decreased in their amplitude but not frequency, and this was normalized by network activity (sIPSCs) (Supplementary Fig. 5b, d). No detectable change was observed in the level of GABA_A receptor subunits or the tonic GABA currents under baseline and GABA-treatment conditions (Supplementary Fig. 5e–g).

These results collectively suggest that Ank2-cKO increases neuronal excitability and firing in the somatosensory cortex, while having moderate or no effect on excitatory and inhibitory synaptic transmission.

### Decreased Kv7.2/3 proteins in the Ank2-cKO brain

To explore the mechanisms underlying the increased neuronal excitability and firing of Ank2-cKO cortical neurons in an unbiased manner, we attempted proteomic analysis of posttranslational modifications (PTMs; mainly phosphorylation) in the Ank2-cKO brain (cortex +

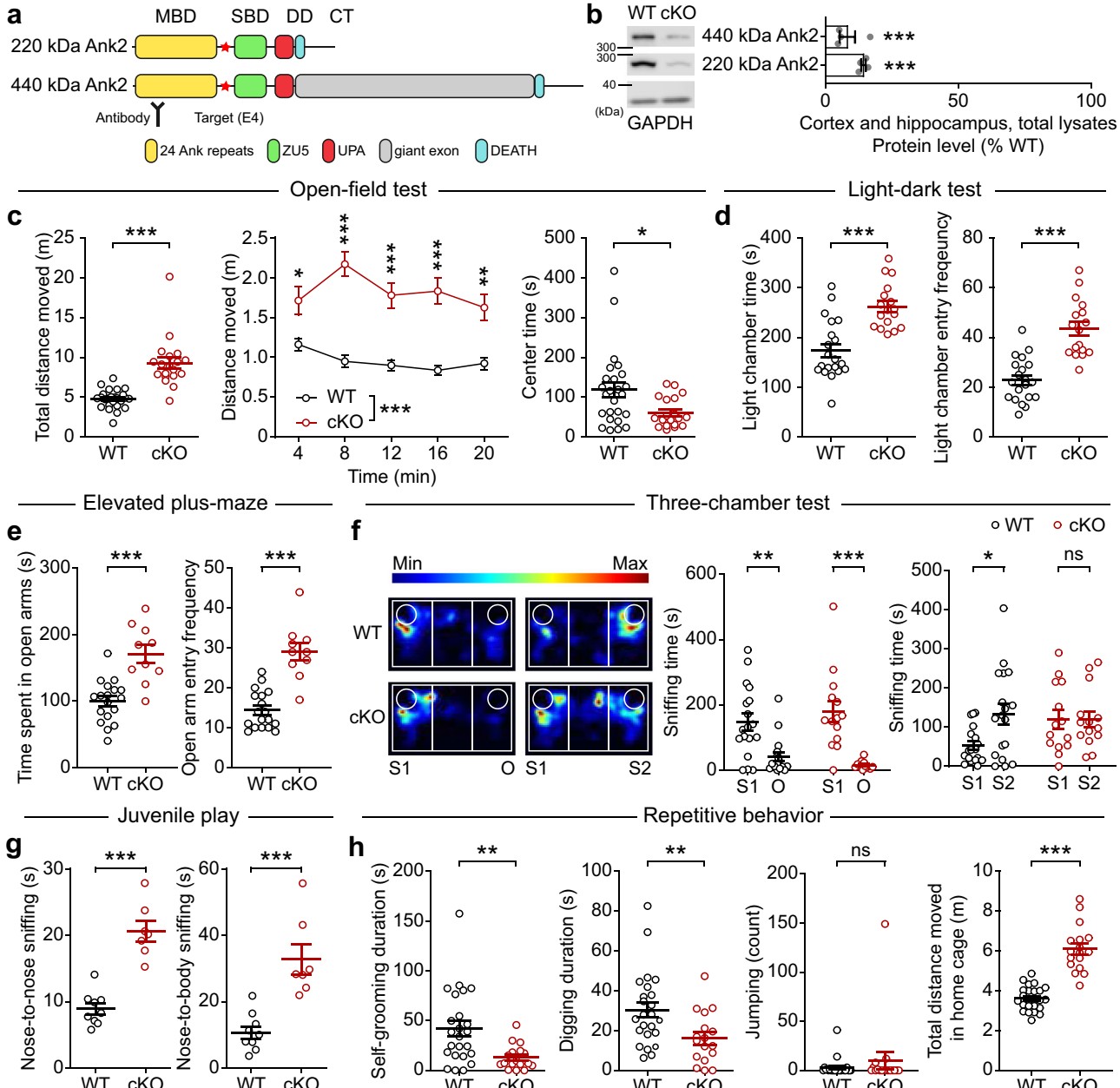

**Fig. 1 | Behavioral deficits in Ank2-cKO mice. a** Domain structures of short and long Ank2 protein variants (220 and 440 kDa, respectively). Note that exon 4 deletion (red star) affects both short and long variants. MBD membrane-binding domain, SBD spectrin-binding domain; linker, a linker for the MBD and SBD domains; giant exon, a domain uniquely present in the long Ank2 variant; DD death domain, CT C-terminal region. **b** Immunoblot analysis of WT and Ank2-cKO cortical and hippocampal lysates (P21). ($n = 4$ [WT, cKO], one sample $t$-test). **c** Hyperactivity of Ank2-cKO mice (P19–21) in the open-field test (100 lux), as shown by distance moved. The reduced time spent in the center region suggests anxiety-like behavior. ($n = 25$ [WT; males and females mixed here and in all other behavioral experiments], 19 [cKO], Mann-Whitney test [total distance moved, center time], two-way repeated-measures/RM-ANOVA with Sidak's test [distance moved]). **d** Anxiolytic-like behavior in Ank2-cKO mice (P19–21) in the light-dark test (600 lux), as shown by light-chamber time/entry. ($n = 20$ [WT], 17 [cKO], Student's $t$-test). **e** Anxiolytic-like behavior in Ank2-cKO mice (P24–25) in the elevated plus-maze

test, as shown by open-arm time/entry. ($n = 18$ [WT], 10 [cKO], Student's $t$-test). **f** Normal social interaction and impaired social novelty recognition in Ank2-cKO mice (P19–20) in the three-chamber test, as shown by sniffing time. S1/S2, novel/ familiar stranger; O, object. ($n = 18$ [WT], 14 [cKO], Mann–Whitney test [WT-S1/O, WT-S1/S2, and cKO-S1/O], Student's $t$-test [cKO-S1/S2]). **g** Abnormally increased social interaction in Ank2-cKO mice (P23–25) in a direct social-interaction test. ($n = 9$ [WT], 7 [cKO], Student's $t$-test). **h** Decreased self-grooming and digging, without a change in jumping, in Ank2-cKO mice (P25–27). Note that hyperactivity is observed here, similar to the open-field hyperactivity. ($n = 24$ [WT], 17 [cKO], Mann-Whitney test [digging/self-grooming]). Source data for uncropped immunoblot images are provided as a Source Data file. The statistical tests involved two-sided analyses, and adjustments were made for multiple comparisons. Data are presented as mean values +/- SEM. $P$ values in figure panels: *$p < 0.05$, **$p < 0.01$, ***$p < 0.001$, ns, not significant.

hippocampus) (Fig. 5a), as these modifications can reflect the complex functional states of proteins and biological pathways[22].

We detected a large number of proteins with significant up- and downregulations of PTMs (up/down-PTM proteins) from Ank2-cKO

frontal and hippocampal brain regions, including 153 peptides from 125 up-PTM proteins and 1203 peptides from 623 down-PTM proteins ($p < 0.05$), which indicates much stronger PTM downregulations relative to upregulations (Supplementary Dataset 2). DAVID analysis of the

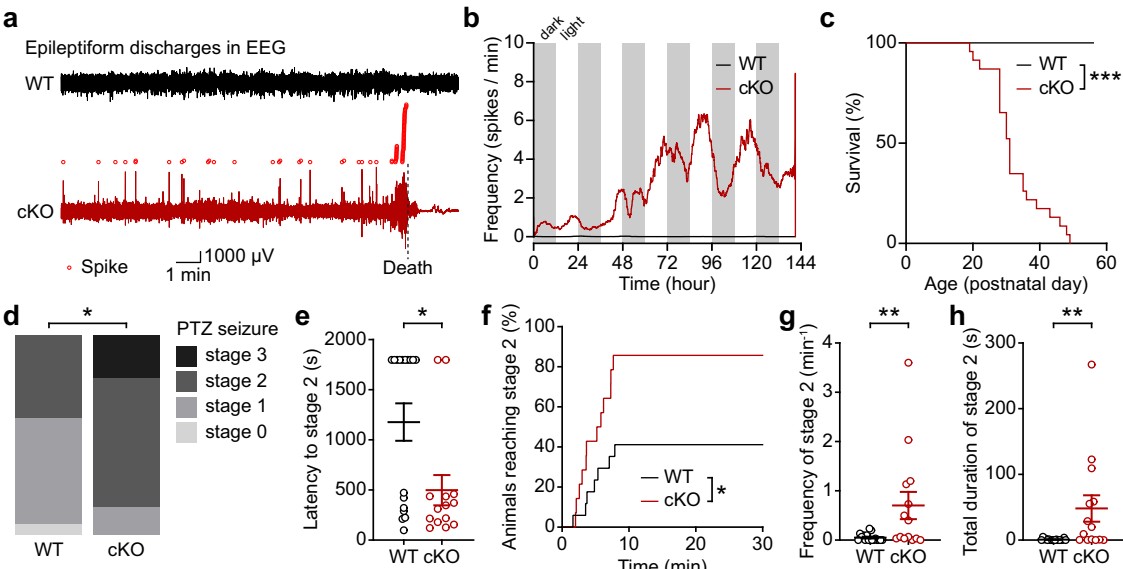

**Fig. 2 | Increased seizures and juvenile death in Ank2-cKO mice. a** Examples of epileptiform spike discharges in Ank2-cKO mice (P35; parietal right lobe) monitored until death, as shown by electroencephalography (EEG). **b** Epileptiform discharges in EEG observed in an Ank2-cKO mouse until death. **c** Juvenile seizure-related death occurring in Ank2-cKO mice during -P20–50. (*n* = 23 [WT], 23 [cKO], Log-rank test). **d–h** Increased pentylenetetrazole (PTZ)-induced seizures in Ank2-cKO mice (P25–27), as shown by final seizure stage (1–4) reached, latency to stage 2 seizure, animals reached stage 2, and frequency and total duration of stage 2. The stage 2 was mainly compared because of its predominance in the mutant mice. (*n* = 17 [WT], 10 [cKO], chi-square test [final seizure stage], Mann-Whitney test [stage 2 latency/duration]). The statistical tests involved two-sided analyses. Data are presented as mean values +/− SEM. *P*-values in figure panels: \**p* < 0.05, \*\**p* < 0.01, \*\*\**p* < 0.001, ns, not significant.

PTM proteins revealed gene ontology (GO) functions in the cellular component domain that are related to neuronal synapses, such as the postsynaptic density and glutamatergic synapse, in both up- and down-PTM proteins (Fig. 5b, c). GO terms such as protein binding and protein kinase binding could also be identified in the molecular function domain. The PTM proteins were also enriched for synaptic proteins in the SynGO database[23], where down-PTM proteins were more strongly overlapped with SynGO proteins in total number than up-PTM proteins, and postsynaptic enrichments were stronger than presynaptic enrichments for both up- and down-PTM proteins (Supplementary Fig. 6).

The PTM proteins with significant p values and strong up/down fold changes (>1.5), indicated in the volcano plots, included those belonging to SynGO proteins and non-SynGO proteins (Fig. 5d, e). Examples of up-PTM SynGO proteins included calcium voltage-gated channel subunit alpha1 A (CACNA1A), glutamate ionotropic receptor kainate type subunit 5 (GRIK5), sodium voltage-gated channel alpha subunit 2 (SCN2A), and adapter protein, phosphotyrosine interacting with PH domain and leucine zipper 1 (APPL1) and those of down-PTM SynGO proteins included neurofilament medium (NFM), microtubule-associated protein 2 (MAP2), Kv3.1, Kv7.2, latrophilin 2, ankyrin repeat and sterile alpha motif domain containing 1B (ANKS1B), and diacylglycerol lipase alpha (DAGLA). Intriguingly, many of the synaptic proteins with downregulated PTMs corresponded to potassium channel subunits or their related proteins; these included Kv1.4, Kv2.1, Kv3.1, Kv3.3, Kv7.2, and Kv7.3 (Fig. 5f).

Of the Kv channels, Kv7.2 and Kv7.3 were particularly interesting because they represent principle subunits of Kv7/KCNQ channels, which are known to regulate neuronal excitability[24–27] and have been implicated in ASD[2,28,29] and epilepsy[30]. The altered Kv7 phosphorylation may stem from changes in phosphorylation or total protein levels. Our proteomic analysis of total (whole-lysate) proteins (not PTM) revealed decreases in Kv7.2 and Kv7.3 levels (Fig. 5g–l; Supplementary Dataset 3). The extents of these changes were 2–3-fold smaller than those observed in PTM levels, suggesting that the observed changes in PTM levels may have involved changes in both total and PTM levels.

Our immunoblot experiments also confirmed that the total (whole-lysate) and synaptosomal levels of Kv7.2 and Kv7.3, but not those of Kv2.1, Kv3.1, or Kv1.2, were decreased in the Ank2-cKO brain (cortex + hippocampus; Fig. 5j). Nav1.2 and Nav1.6 sodium channel subunits, known to interact with Ank3 and regulate neuronal excitability[31,32], were not and moderately decreased in synaptosomal and total samples, respectively (Supplementary Fig. 7a).

How might Ank2 deletion decrease Kv7 protein levels? The specific phosphorylation sites that we identified in Kv7 channels (Kv7.2-Ser-410/438 and Kv7.3-Ser-457), which have not been reported previously, might play a role. However, phospho-mimic and non-phosphorylatable point mutations introduced to Kv7.2/3 (Kv7.2-Ser-410/438-Glu/Ala and Kv7.3-Ser-457-Glu/Ala) did not alter protein stability in cultured neurons (Supplementary Fig. 7b).

Ank2 may control Kv7 stability either directly through the interaction with Kv7 or indirectly through an indirect association with Ank3. However, our coimmunoprecipitation experiments did not support the biochemical association between Ank2 and Kv7.2/3, or Ank2 and Ank3, while the known Ank3-Kv7.2/3 interaction could be reproduced (Supplementary Fig. 7c, d). Interestingly, cultured Ank2-cKO neurons showed substantially reduced surface levels of Kv7.2 (Fig. 5k), suggesting that Ank2 deletion destabilizes Kv7 at the membrane surface.

These results suggest that Ank2 cKO decreases the total and surface levels of Kv7 proteins and alters phosphorylation levels of various brain proteins, including Kv7.2/3.

**Altered total and synaptic proteomes in Ank2-cKO mice**

Kv7.2/3 proteins are present in neuronal axons, dendrites, synapses, and nodes of Ranvier, and function to regulate AP generation/properties/propagation and dendritic excitability[25,33–37]. They are also enriched in the axon initial segment (AIS)[26,31,35], which regulates AP generation and properties[14,15]. Hence, we first examined whether the synaptic localization of Kv7.2/3 is compromised in the Ank2-cKO brain.

Kv7.3 was detected at small fractions (-5–10%) of both excitatory and inhibitory synapses in cultured cortical neurons, but more at excitatory synapses than inhibitory synapses (Supplementary Fig. 8a).

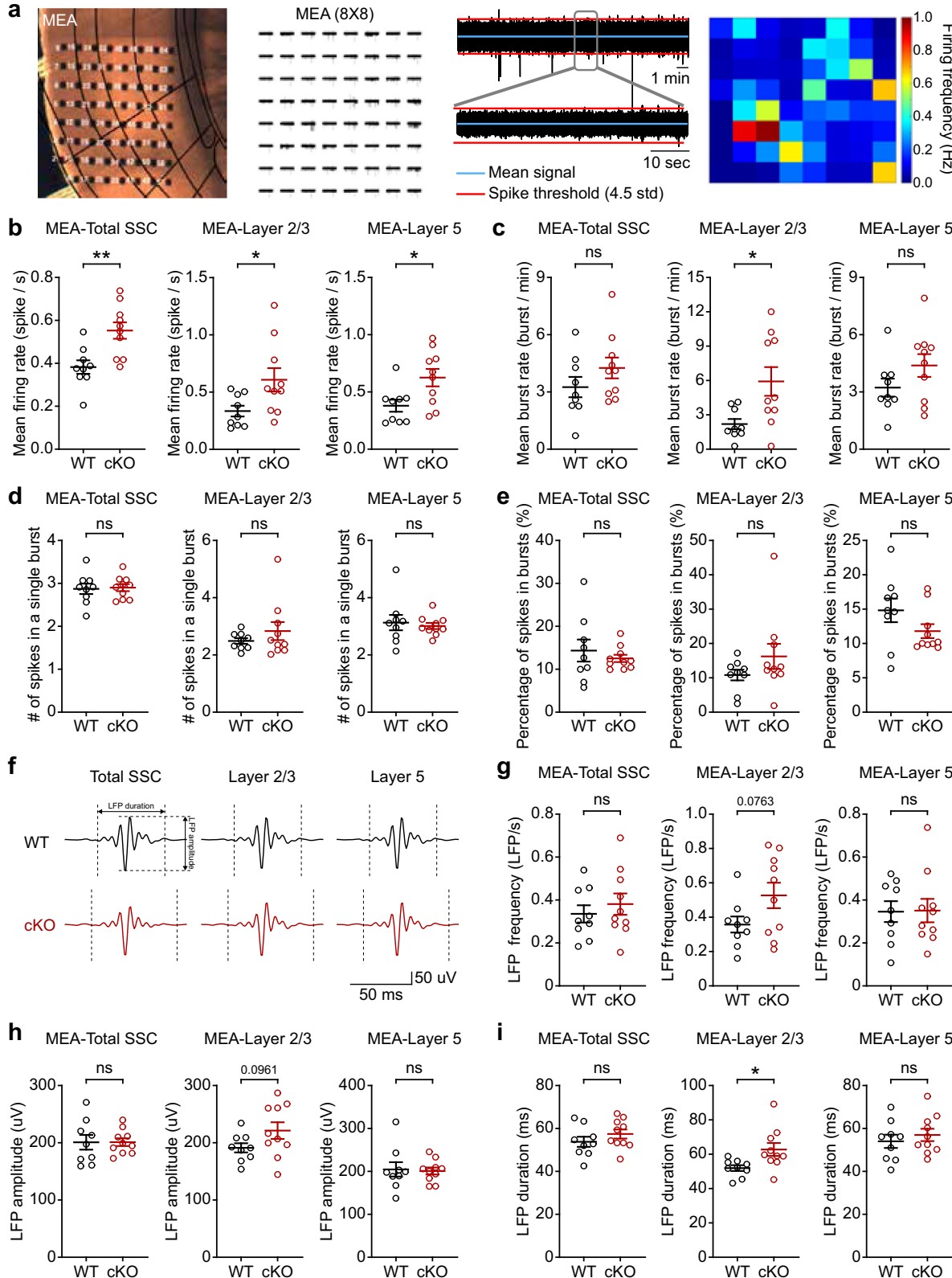

In addition, Ank2-cKO neurons displayed moderately decreased synaptic localization of Kv7.3 at excitatory but not inhibitory synapses. Kv7.2 staining could not be performed for the lack of suitable antibodies. The reduced synaptic localization of Kv7.3 suggests that mutant excitatory synapses underwent chemical alterations. We thus attempted a proteomic analysis of Ank2-cKO crude synaptosomes and compared the results with those from the aforementioned total proteome analysis.

The numbers of proteins differentially expressed (DEPs) in WT and Ank2-cKO mice were larger in the synaptic proteome than in the total proteome (519 [251 up and 268 down] in synaptosomal and 240 [93 up and 147 down] in total) (Supplementary Fig. 9a,b; Supplementary Dataset 4). Kv7.2 and Kv7.3 observed in total DEPs (Fig. 5i) were not detected in synaptosomal DEPs (Supplementary Fig. 9c). Notably, Ank2 was found to be downregulated (-2.6-folds) in both synaptosomal and total DEPs, suggesting that Ank2 localizes to synapses and may

**Fig. 3 | Increased cortical excitability in the Ank2-cKO brain. a** Examples of neuronal firing in the somatosensory cortex (SSC) in brain slices from Ank2-cKO mice, as measured by multielectrode array (MEA; 64 electrodes in 8 × 8 arrays; cutoff for significant firing > 4.5 x standard deviation). **b, c** Increased mean firing rate and mean burst rate (inter-spike-interval <10 ms) in the Ank2-cKO SSC (P19–22), as measured by MEA. (*n* = 9 slices/3 mice [WT], 10, 3 [cKO], Student's *t*-test [total], Welch's test [layer 2/3], Mann–Whitney test [layer 5]). **d** Unaltered number of spikes in a single burst in the Ank2-cKO SSC (P19–22) measured by MEA. (*n* = 9, 3 [WT], 10, 3 [cKO], Student's t-test [total], Mann–Whitney test [layer 2/3 and 5]). **e** Unaltered percentage of spikes in bursts in the Ank2-cKO SSC (P19–22) measured

by MEA. (*n* = 9, 3 [WT], 10, 3 [cKO], Welch's test [total], Mann-Whitney test [layer 2/3 and 5]). **f** Diagram showing examples of local field potentials (LFPs) and depicting parameters of LFP (length and amplitude). **g–i** Increased duration but normal frequency and amplitude of LFPs in layer 2/3 but not layer 5 in the Ank2-cKO SSC (P19–22) measured by MEA. (*n* = 9, 3 [WT], 10, 3 [cKO], Student's *t*-test [LFP frequency-layer 2/3, amplitude-total and layer 2/3, duration-total and layer 5], Mann–Whitney test [frequency/amplitude-layer 5], Welch's test [duration-layer 2/3]). The statistical tests involved two-sided analyses. Data are presented as mean values +/- SEM. *P*-values in figure panels: **p* < 0.05, ***p* < 0.01, ****p* < 0.001, ns, not significant.

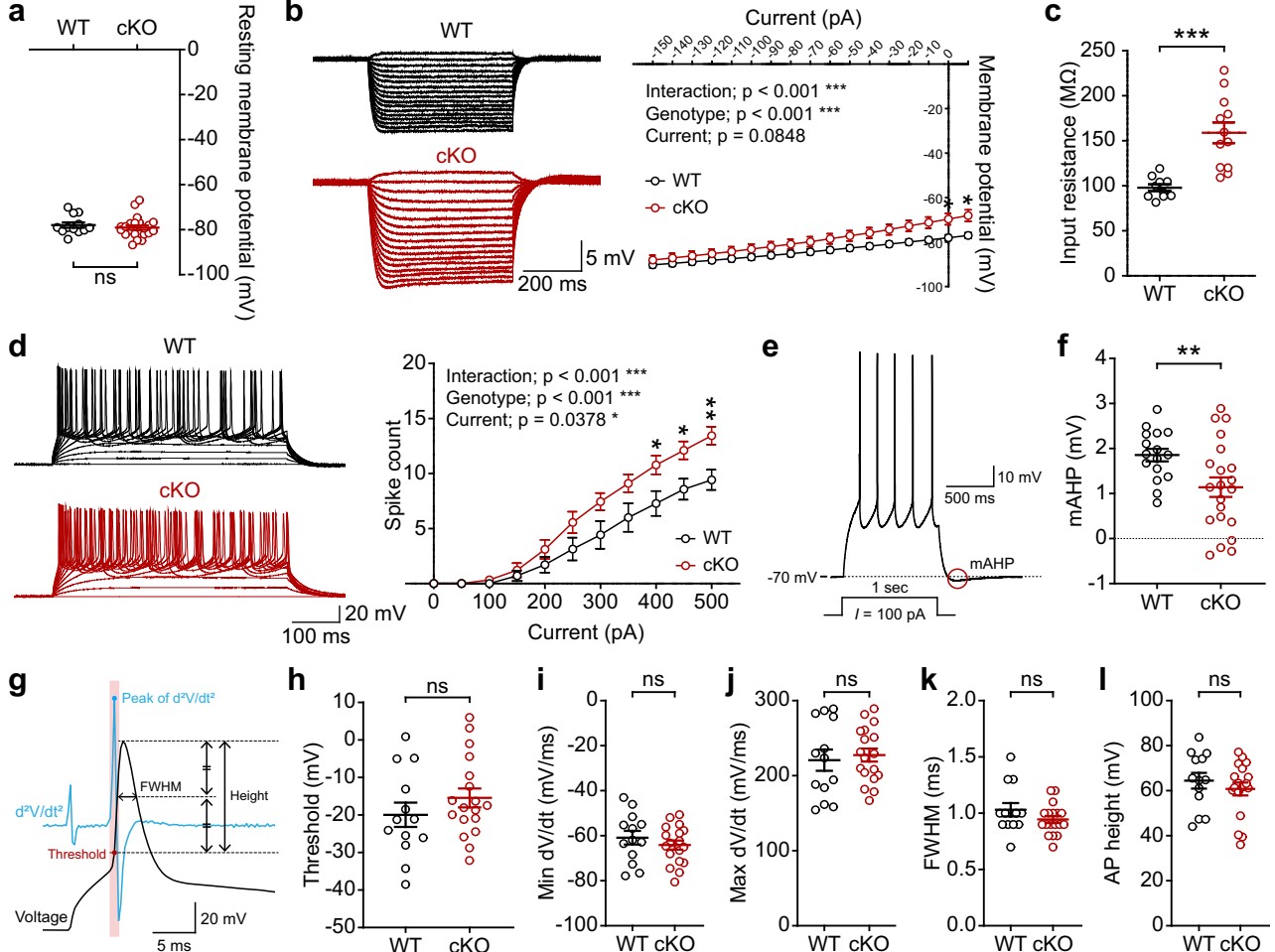

**Fig. 4 | Increased neuronal excitability in the Ank2-cKO cortex. a** Normal resting membrane potential in Ank2-cKO SSC layer 2/3 pyramidal neurons (P19–22). (*n* = 12 neurons/8 mice [WT], 22,8 [cKO], Mann–Whitney test). **b, c** Increased input resistance in Ank2-cKO SSC layer 2/3 neurons (P19–22). (*n* = 10, 3 [WT], 12, 3 [cKO], two-way RM-ANOVA with Sidak's test and Student's *t*-test). **d** Increased current-firing curve slope in Ank2-cKO SSC layer 2/3 neurons (P19–22). (*n* = 7, 3 [WT], 9, 3 [cKO], two-way RM-ANOVA with Sidak's test). **e** Diagram explaining mAHP. **f** Decreased mAHP amplitude in Ank2-cKO SSC layer 2/3 neurons (P19–22). (*n* = 16, 5 [WT], 21, 5 [cKO], Welch's test). **g** Diagram explaining AP threshold and AP shape-related

parameters. **h–l** Normal AP threshold and AP shape-related parameters in Ank2-cKO SSC layer 2/3 neurons (P19–22), as indicated by AP-threshold voltage and the height, width, and time-dependent voltage changes (dV/dt; maximal/depolarizing and minimal/repolarizing) of APs. (*n* = 13, 9 [WT], 18, 11 [cKO], Student's *t*-test [AP threshold and Max dV/dt and min dV/dt], Mann–Whitney test [FWHM and amplitude]). The statistical tests involved two-sided analyses, and adjustments were made for multiple comparisons. Data are presented as mean values +/− SEM. *P*-values in figure panels: **p* < 0.05, ***p* < 0.01, ****p* < 0.001, ns, not significant.

regulate synaptic functions, although the synaptic localization of Ank2 could not be directly tested for the lack of suitable antibodies.

Functionally, synaptosomal DEPs, in particular those that are upregulated, were enriched for synapse-related gene ontology terms in the DAVID analysis whereas downregulated synaptosomal DEP or total DEPs (up and downregulated) did not (Supplementary Fig. 9d, e). Notably, upregulated total DEPs were enriched for protein phosphatase-related terms, which may explain the strong PTM downregulations mentioned above. In SynGO analyses, a larger

number of synaptosomal DEPs belonged to SynGO proteins, as compared with that for total DEPs (Supplementary Fig. 10). In addition, up- but not down-regulated DEPs were more strongly enriched for SynGO proteins in both synaptosomal and total DEPs.

Additional analyses of the DEPs using GSEA[38] (termed PSEA hereafter) revealed strong and positive enrichments of synaptosomal DEPs for synapse-related gene sets, as shown by top-five gene sets and gene-set clusters generated using Cytoscape EnrichmentMap App[39] (Supplementary Fig. 11; Supplementary Dataset 5). Similar enrichment

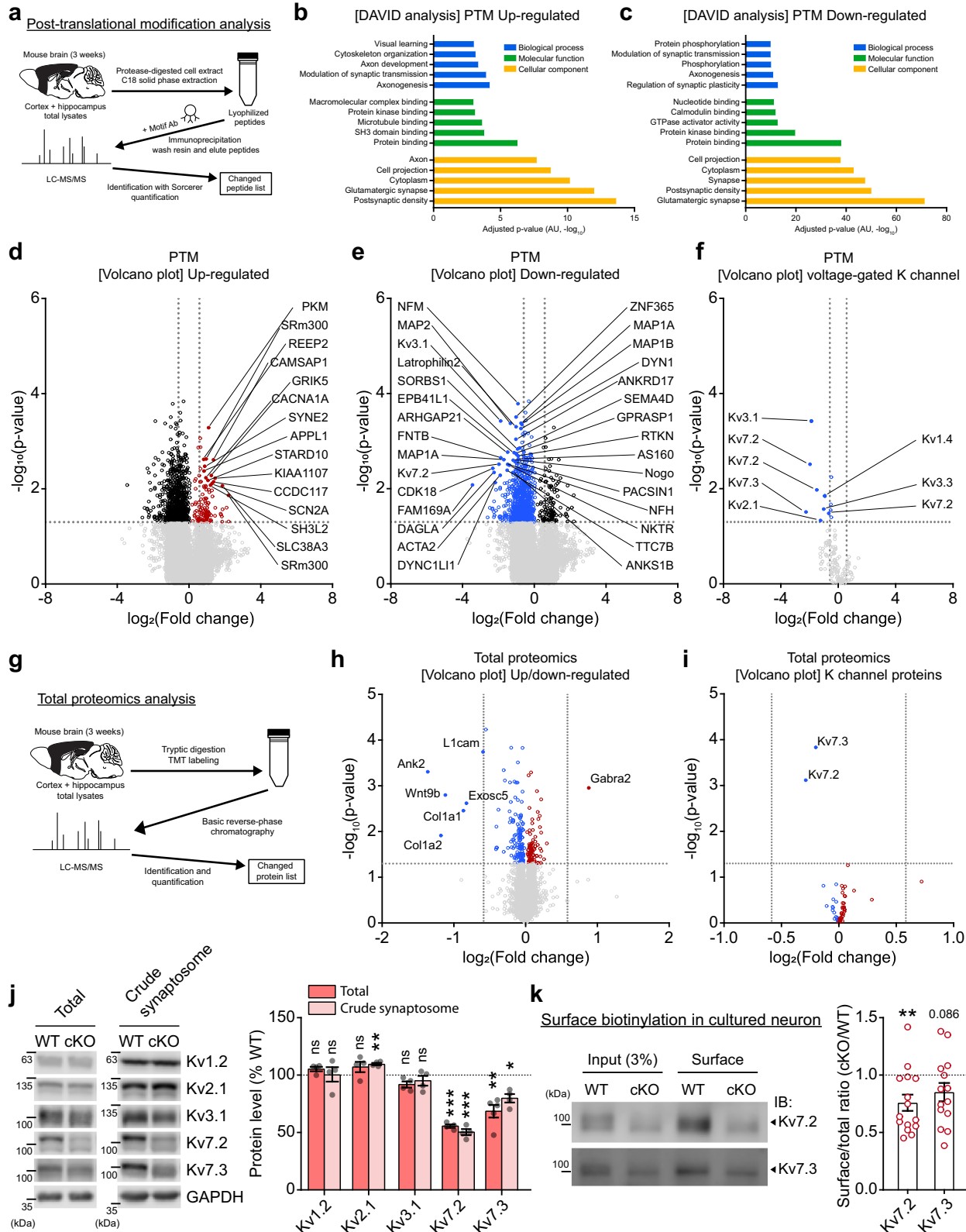

patterns were observed in total DEPs, though to a lesser extent than in synaptosomal DEPs.

An additional PSEA for ASD-risk/related gene sets revealed patterns that are opposite to those observed in ASD[40] (termed reverse-ASD) in synaptosomal DEPs but not in total DEPs (Supplementary Fig. 12a–c; Supplementary Datasets 5 and 6). It included positive enrichments for ASD-risk gene sets (i.e., SFARI and FMRP-target gene

sets) and ASD-related single-cell-specific gene sets (i.e., neuron- and oligodendrocyte-related gene sets).

When the overlaps between DEPs and ASD-risk proteins were analyzed, upregulated synaptosomal DEPs overlapped more strongly (~2.8-folds) with ASD-risk proteins relative to downregulated synaptosomal DEPs, as shown by Venn diagrams and volcano plots (Supplementary Fig. 12d–h). In contrast, downregulated total DEPs

**Fig. 5 | Decreased Kv7.2/3 proteins in the Ank2-cKO brain. a** Diagram depicting procedures of PTM (posttranslational modification) analyses for samples from WT and Ank2-cKO cortex and hippocampus; P19–22 (*n* = 3 groups from 9 mice [3 mice/group] [WT, cKO]). **b, c** DAVID gene ontology (GO) analyses of biological functions for the proteins with up- and downregulated PTM levels (PTM proteins). **d–f** Volcano plots highlighting the PTM proteins with significant fold changes (>1.5) and *p*-values (<0.05, Welch's *t*-test) and the presence of potassium channels in the down-PTM proteins (fold change > 1.5 and *p* < 0.05 [Welch's *t*-test]). **g** Diagram depicting procedures of proteomic analysis of total (whole-lysate) proteins in Ank2-cKO mice (cortex + hippocampus; P19–22). **h, i** Volcano plots showing the differentially expressed proteins (DEPs) derived from the analysis of total (whole-lysate) proteins in Ank2-cKO mice (cortex and hippocampus; P19–22) with significant fold changes (>1.5) and *p*-values (<0.05) and the presence of Kv7.2 and Kv7.3 in the downregulated total DEPs (*p* < 0.05 [Welch's *t*-test] but not fold change > 1.5). (*n* = 3 mice [WT, cKO]). **j** Decreased total and crude synaptosomal levels of Kv7.2 and Kv7.3 in Ank2-cKO brains (cortex + hippocampus; P19–23). Note that the levels of Kv2.1, Kv3.1, and Kv1.2 (unrelated control) were not changed. (*n* = 4 mice [WT, cKO] except for Kv7.3 in total lysates (*n* = 5 mice [WT, cKO]), one sample *t*-test). **k** Decreased surface levels of Kv7.2 and Kv7.3 in cultured cortical neurons (DIV 17). (*n* = 15 dishes [5 + 5 + 5] from 3 biologically independent experiments [Kv7.2] and 14 dishes [5 + 5 + 4] from 3 biologically independent experiments [Kv7.3], one sample t-test). Source data for uncropped immunoblot images are provided as a Source Data file. The statistical tests involved two-sided analyses. Data are presented as mean values +/– SEM. *P*-values in figure panels: *\**p* < 0.05, **\*\**p* < 0.01, ***\*\*\**p* < 0.001, ns, not significant.

overlapped more strongly (~2.1-folds) with ASD-risk proteins relative to upregulated total DEPs. Intriguingly, downregulated PTM proteins overlapped much more strongly (~7.4-folds) with ASD-risk proteins relative to upregulated PTM proteins. Many of the overlapping proteins were also SynGO proteins.

These results collectively suggest that Ank2 cKO leads to (1) a greater number of DEPs in the synaptosomal proteome relative to the total proteome, (2) stronger upregulations of synapse-related genes in the synaptosomal proteome relative to the total proteome, (3) a reverse-ASD pattern in the synaptosomal but not in the total proteome, and (4) differential up/downregulations of ASD-risk proteins in total and synaptosomal proteomes.

### Decreased Kv7 density in the enlengthened Ank2-cKO AIS

We next tested if there are any alterations in the morphology and function of the AIS where Kv7.2/3 are strongly enriched[26,31,35]. An AAV-Cre-mediated acute knockout of Ank2 in cultured *Ank2fl/fl* cortical neurons increased the area of the AIS, which was marked by Ank3, but decreased the mean intensity of the signal in the AIS, such that the total Ank3 signal in the AIS was unchanged (Fig. 6a, b). Kv7.3, which is enriched in the AIS by its direct interaction with Ank3[31,41], was similarly decreased in its mean intensity, resulting in unchanged total levels in the enlengthened mutant AIS (Fig. 6c).

In addition, immunostaining and three-dimensional imaging and quantification using brain tissues physically expanded by the epitope-preserving magnified analysis of proteome (eMAP) technology (see Methods for details) (Fig. 6d) indicated that the length and area of Ank3-labeled AIS were increased in the Ank2-cKO somatosensory cortex (Fig. 6e–g). In contrast, the length of the node of Ranvier, where Kv7 proteins are also localized[35,37], was not changed (Supplementary Fig. 8b). These in vitro and in vivo results suggest that the decreased mean intensity of Kv7.2/3 and Ank3 may decrease the current density of Kv7.2/3 in the mutant AIS, thereby decreasing the amplitude of post-burst/sustained firing mAHP. In addition, given that Ank2 and spectrins (αII and βII) together set the boundary of the AIS in the proximal axon[16], Ank2 deletion may increase the length of the AIS and thereby decrease Kv7 channel density.

To further explore Ank2-cKO-induced Kv7.2/3 dysfunctions, we next directly measured Kv7.2/3-mediated neuronal excitation induced by the Kv7 antagonist, XE991. WT SSC layer 2/3 pyramidal slices exhibited the expected increase in neuronal excitability upon XE991 treatment, as shown by the current-firing curve; in contrast, Ank2-cKO neurons did not show any significant XE991-induced excitation (Fig. 6h, i). Therefore, Ank2-cKO neurons with decreased Kv7.2/3 levels and increased baseline neuronal excitability are less susceptible to XE991-induced neuronal excitation relative to WT neurons.

These results collectively suggest that Ank2 cKO decreased the levels of Kv7.2/3 and the AIS intensity of Kv7.2/3 and increased excitability and firing in cortical neurons.

### Kv7 activation rescues neuronal excitability and juvenile death in Ank2-cKO mice

We next tested if the FDA-approved Kv7 agonist, retigabine[42], could rescue the increased neuronal excitability and firing in Ank2-cKO cortical neurons and the juvenile seizure-related death in Ank2-cKO mice. Retigabine treatment was started at ~P16-17; this represented a time point immediately before the onset of juvenile seizure-related death and within the period during which Ank2, Kv7.2, and Kv7.3 protein levels were sharply increased (Supplementary Fig. 13a–c). Around this time (~P14), Ank2 and Kv7.2/3 protein levels began to decline synchronously in Ank2-cKO mice without changes in their mRNA levels (Supplementary Fig. 13d–i).

Acute retigabine treatment rescued the increased excitability and firing in Ank2-cKO cortical neurons to WT levels without affecting those in WT neurons, as indicated by current-firing curves in slice preparations (Fig. 7a, b). Retigabine also rescued the reduced mAHP amplitude in the mutant neurons without affecting that in WT neurons (Fig. 7c, d). Retigabine, however, did not alter the resting membrane potential, input resistance, AP threshold, or AP shape-related parameters (Fig. 7e, f; Supplementary Fig. 14).

More importantly, retigabine treatment improved the juvenile seizure-related death in Ank2-cKO mice, enabling ~20% of the mutant mice to survive until P80, whereas it had no effect on WT mouse survival (Fig. 8a, b). Retigabine effects vanished when the medication was withdrawn at P42 (26 days after the start of therapy), which led to seizure-related death (Fig. 8c, d), suggesting that retigabine has no long-lasting effects. Retigabine, however, did not rescue the hyperactivity or anxiety-like behavior of Ank2-cKO mice in the open-field test and had no effect on the body weight (Fig. 8e–g; Supplementary Fig. 15a).

The lack of a rescue effect on open-field hyperactivity and anxiety-like behavior may be due to the pharmacological effects of chronic retigabine treatment being intermittent in nature. In additional experiments for acute treatment, retigabine could rescue open-field hyperactivity with no effect on anxiety-like behavior, although the rescue effect declined rapidly during the 10 min of behavioral monitoring (Fig. 8h–k), likely reflecting drug metabolism. The acute retigabine treatment, however, did not rescue repetitive behavior (self-grooming and digging) or juvenile play (Supplementary Fig. 15b–f). The lack of baseline differences in open-field center time and digging in the mutant mice might reflect pain-related responses to the acute injection.

These results collectively suggest that retigabine-induced Kv7 activation in Ank2-cKO mice improves hyperactivity and juvenile seizure-related death and normalizes neuronal excitability and firing in mutant neurons mainly by correcting the mAHP amplitude.

## Discussion

Here, we investigated the in vivo functions of Ank2 and found that cortical Ank2 deletion in mice leads to ASD-related behavioral deficits

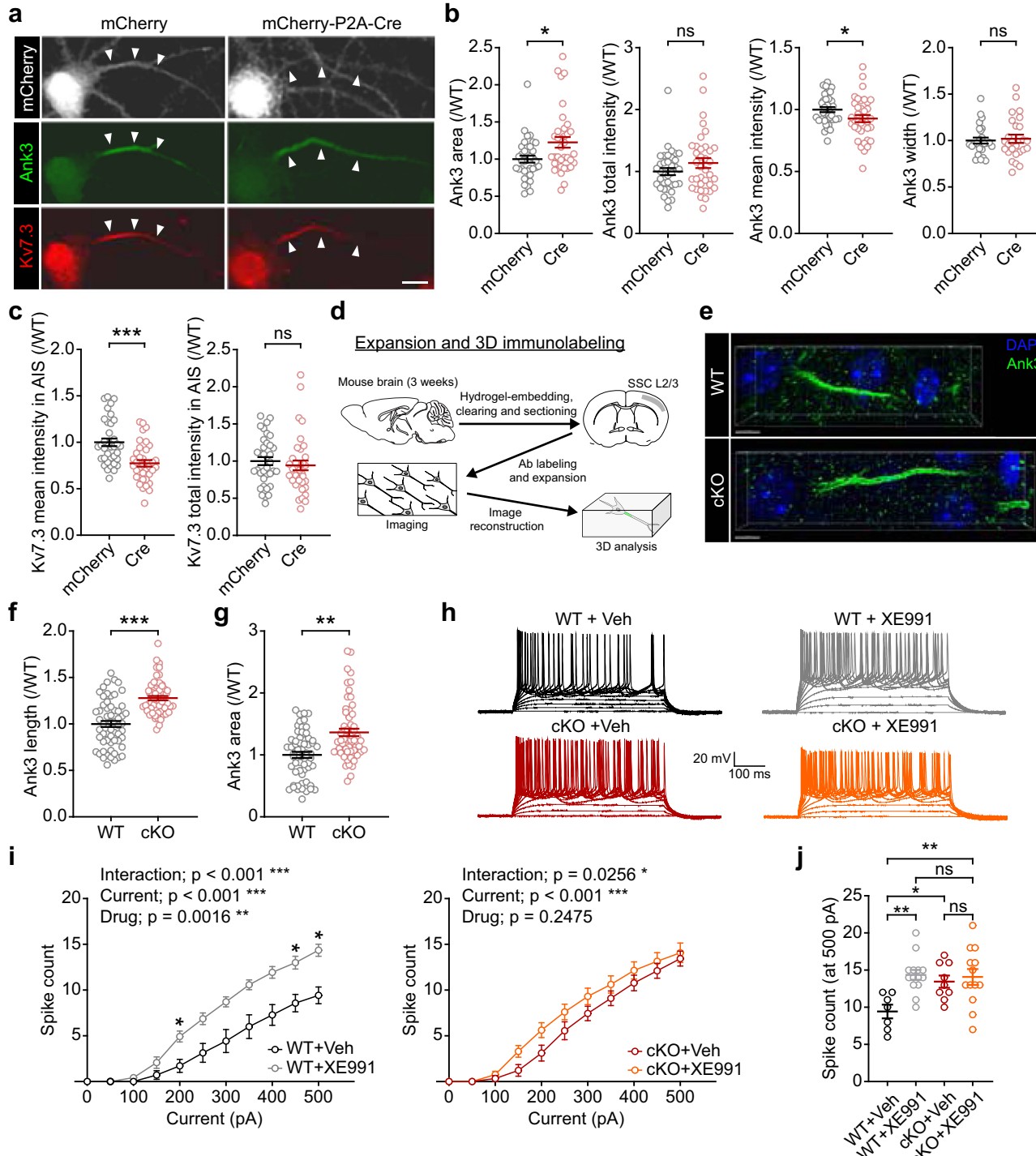

**Fig. 6 | Decreased Kv7.3 density in the enlengthened AIS and decreased Kv7 function in the Ank2-cKO brain. a–c** Increased AIS area (marked by Ank3; arrows) in cultured hippocampal *Ank2^fl/fl* neurons infected with AAV-Cre (DIV 4 ~ 10–14 ~ 28), and decreased mean intensities of Kv7.3 and Ank3 in the AIS area; these changes together yield normal total intensities of Kv7.3 and Ank3. Note that the width of Ank3 was not changed. (*n* = 36 neurons from three independent experiments [WT], 36 [cKO], Welch's test [Ank3 mean intensity], Mann–Whitney test [Ank3 area/total intensity; Kv7.3 mean/total intensity in AIS]). Scale bar, 10 μm. **d–g** Increased AIS length and area (marked by Ank3) in the Ank2-cKO somatosensory cortex (P19–23), as determined by three-dimensional imaging and quantification of expanded brain tissues. (*n* = 60 neurons from 4 mice [WT], 60, 4 [cKO], Student's t-test [Ank3 length], Welch's *t*-test [Ank3 area]). Scale bar, 4 μm. **h–j** The Kv7.2/3 antagonist, XE991, increases the slope of the current-spike curve in WT, but not Ank2-cKO, SSC layer 2/3 pyramidal neurons (P19–23). (*n* = 7 neurons from 3 mice [WT], 14,4 [WT + XE991], 9,3 [cKO], 13,5 [cKO+XE991], two-way RM-ANOVA with Sidak's test [i] and with Tukey's test [j]). The statistical tests involved two-sided analyses, and adjustments were made for multiple comparisons. Data are presented as mean values +/− SEM. *P*-values in figure panels: *p < 0.05, **p < 0.01, ***p < 0.001, ns, not significant.

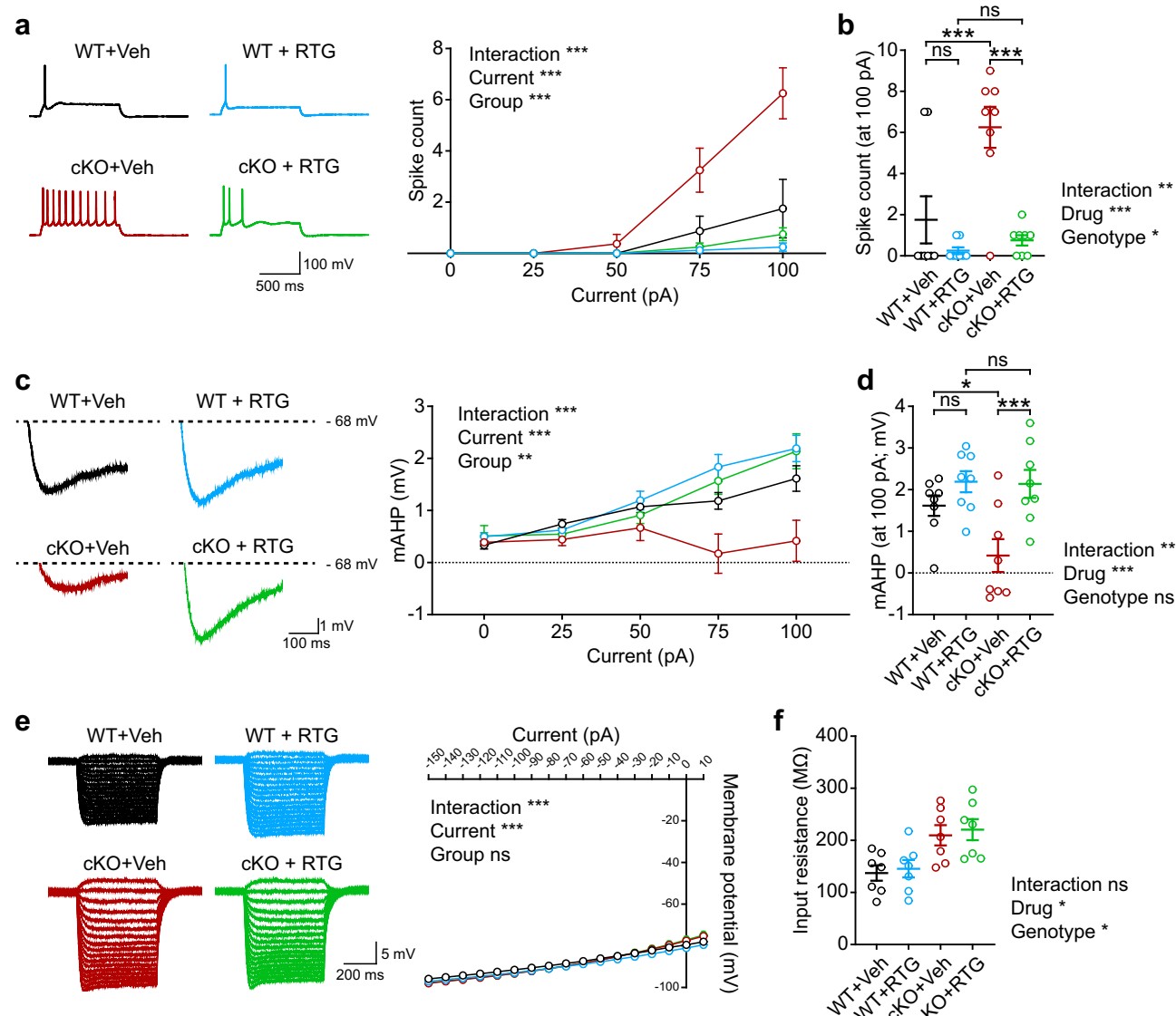

**Fig. 7 | Kv7 activation rescues neuronal excitability in Ank2-cKO mice.**
**a**, **b** Retigabine treatment (from P16–17) rescues the increased neuronal excitability and firing in Ank2-cKO SSC layer 2/3 pyramidal neurons (P19–23) without affecting WT neurons, as shown by current-firing curves. (*n* = 8 neurons from 4 mice [WT-Veh], 8,4 [WT-RTG], 8,4 [cKO-Veh], 8,4 [cKO-RTG], two-way RM-ANOVA with Tukey's test [a], two-way RM-ANOVA with Sidak's test [b]). **c** and **d** Retigabine treatment (from P16–17) rescues the decreased mAHP (medium after-hyperpolarization) amplitude in Ank2-cKO SSC layer 2/3 neurons (P19–23) without affecting that of WT neurons. (*n* = 8, 4 [WT], 8,4 [WT-RTG], 8,4 [cKO], 8,4 [cKO-

RTG], two-way RM-ANOVA with Tukey's test [c], two-way RM-ANOVA with Sidak's test [d]). **e**, **f** Retigabine treatment (from P16–17) does not affect input resistance in Ank2-cKO or WT SSC layer 2/3 neurons (P19–23). (*n* = 7,4 [WT], 7,4 [WT-RTG], 7,4 [cKO], 7,4 [cKO-RTG], two-way RM-ANOVA with Tukey's test [e], two-way RM-ANOVA with Sidak's test [f]). The statistical tests involved two-sided analyses, and adjustments were made for multiple comparisons. Data are presented as mean values +/- SEM. *P*-values in figure panels: *$p < 0.05$, **$p < 0.01$, ***$p < 0.001$, ns, not significant.

associated with juvenile seizure-related death and increased neuronal excitability and firing, through mechanisms that include AIS enlengthening and Kv7.2/3 dysfunction.

Ank2 cKO leads to behavioral deficits, including hyperactivity, altered anxiety-like behavior, excessive social interaction, impaired social novelty recognition, suppressed repetitive behavior, and juvenile seizure-related death (Figs. 1 and 2). An early study on *Ank2* homozygous mutant mice targeted both the short and long protein variants (220 and 440 kDa), which resulted in neonatal mortality and made behavioral assessment difficult[11]. A more recent study characterized three *Ank2*-mutant mouse lines (two for 440 kDa-only deletion and one for 220 and 440 kDa deletion)[12], with the latter (lacking exon 22) being comparable in design to our *Ank2*[+/−] mice (not Ank2-cKO) lacking exon 4. Our *Ank2*[+/−] mice (adult male) show largely normal

behaviors, except for modest anxiety-like behavior (decreased open-field center time) (Supplementary Fig. 2). Our female *Ank2*[+/−] mice (adult) also display largely normal behaviors except for decreased self-grooming (Supplementary Fig. 3). The previously reported *Ank2*[+/−] mice (exon 22; male) show impaired urine marking in adult males and suppressed ultrasonic vocalizations in juvenile males[12] (results from all published *Ank2*-mutant mice are summarized in Supplementary Dataset 7). Therefore, Ank2 exon 4- and 22-mutant mouse lines seem to show distinct behavioral phenotypes in the tests that do not overlap between the two studies, which needs additional characterization. In addition, a more recent study on *Ank2*[+/−] mice lacking only the 440 kDa Ank2 variant (P2580fs line)[12] did not show seizure activity[43], simialr to our *Ank2*[+/−] mice (Supplementary Figs. 2 and 3). Therefore, the stronger decline of Ank2 proteins (~10% of WT levels for both 440- and 220-kDa

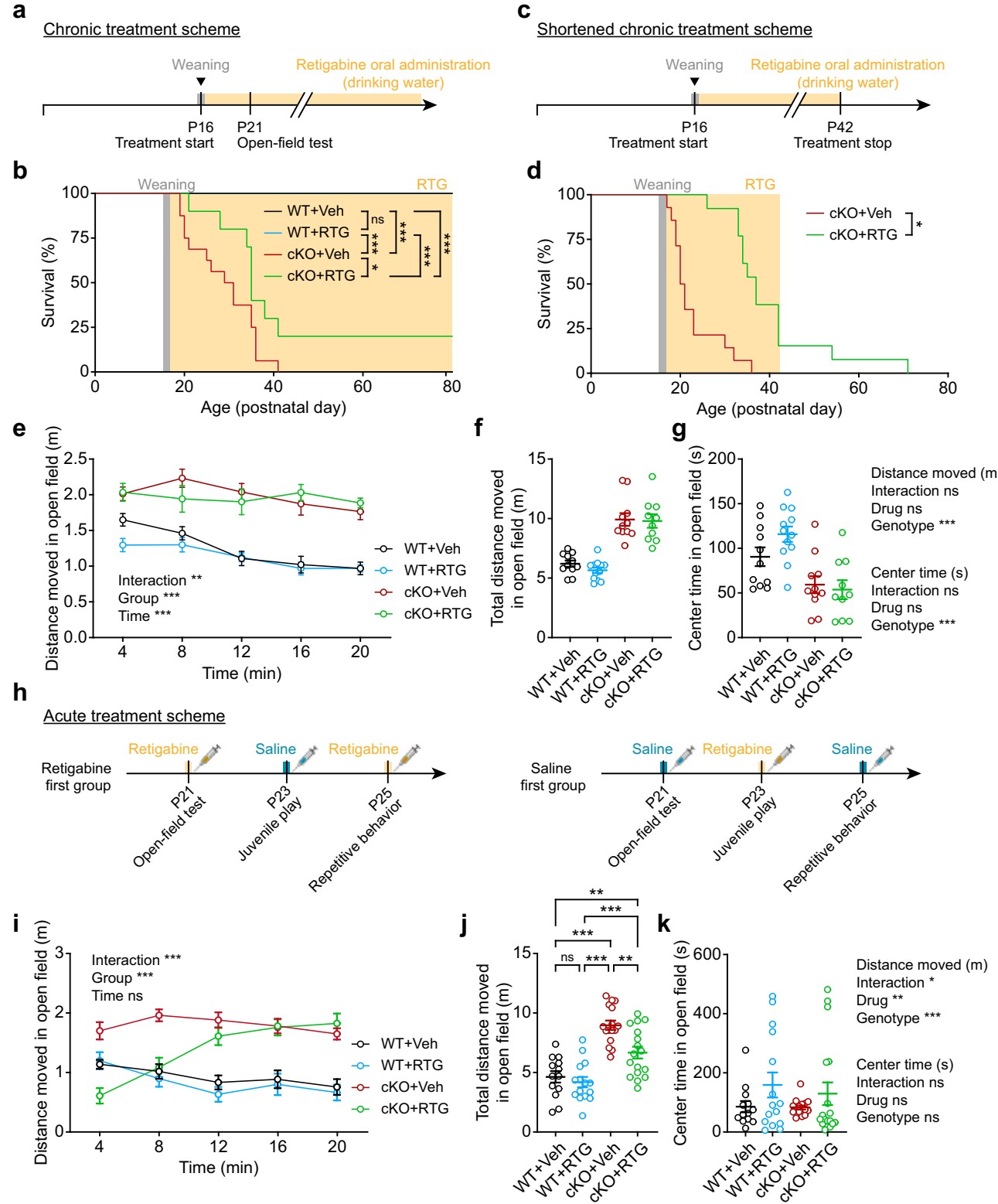

variants) rather than the type of Ank2 variants lost seem to cause the seizure phenotype in Ank2-mutant mice.

The two mouse lines that lack the 440- but not 220-kDa variant in the previous study mentioned above show behavioral changes in locomotor, cognitive, and social domains[12]. However, none of these mice show seizure-related death observed in our Ank2-cKO mice. This difference may reflect that our mice exhibited stronger decreases in both Ank2 variants (440 + 220 kDa) in the cortical/hippocampal areas,

to 10–20% of those seen in WT mice. We speculate that the strong seizure-related death may be relevant to the epilepsy comorbidity in ASD and to *ANK2* human mutations associated with seizure, epilepsy, and sudden death (deciphergenomics/org)[44].

The MEA and patch-clamp results indicate that Ank2 cKO increases neuronal excitability and firing through changes in multiple excitability/AP-related parameters: The input resistance was increased, and the mAHP amplitude was decreased, although there was no

**Fig. 8 | Kv7 activation rescues juvenile seizure-related death and open-field hyperactivity in Ank2-cKO mice. a, b** Chronic retigabine treatment (from P16–17) improves juvenile seizure-related death of Ank2-cKO mice without affecting WT mice. (n = 11 mice [WT-Veh/vehicle], 12 [WT-RTG/retigabine], 16 [cKO-Veh], 10 [cKO-RTG], Log-rank test). **c, d** Cessation of chronic retigabine treatment at around ~P42 (~26 days after the initiation of retigabine treatment) eliminates the treatment effect of retigabine on juvenile seizure-related death in Ank2-cKO mice. (n = 14 mice [cKO-Veh], 13 [cKO-RTG], Log-rank test). **e–g** Chronic retigabine treatment (from P16–17) does not improve hyperactivity or anxiety-like behavior (center time) of Ank2-cKO mice in the open-field test. (n = 11 mice [WT-Veh], 12 [WT-RTG], 11 [cKO-Veh], 10 [cKO-RTG], two-way ANOVA with Tukey test). **h–k** Acute retigabine treatment (5 mg/kg) at P21/22 improves hyperactivity but not anxiety-like behavior (center time) of Ank2-cKO mice in the open-field test. Drug treatment groups were divided into two (retigabine/saline first) to minimize cross-treatment effects. (n = 13 mice [WT-Veh], 15 [WT-RTG], 16 [cKO-Veh], 17 [cKO-RTG], two-way ANOVA with Tukey's test [distance moved_4min], Sidak's test [distance moved_total]). The statistical tests involved two-sided analyses, and adjustments were made for multiple comparisons. Data are presented as mean values +/− SEM. P-values in figure panels: *p < 0.05, **p < 0.01, ***p < 0.001, ns, not significant.

change in the resting membrane potential, AP threshold, or shape-related parameters (Figs. 3 and 4). These changes accompanied decreases in the total and surface levels and functions of Kv7.2/3 as well as increased length of the AIS and decreased density of Kv7 channels in the enlengthened AIS (Figs. 5 and 6). Importantly, Kv7 activation by retigabine improved juvenile seizure-related death and hyperactivity in Ank2-cKO mice by normalizing mAHP amplitude without affecting the resting membrane potential, input resistance, or AP threshold (Figs. 7 and 8). These results collectively suggest that Ank2 cKO increases neuronal excitability by altering excitability/AP-related parameters that act upstream of AP generation (input resistance) or after an AP (mAHP amplitude), but not during an AP (AP shape-related parameters).

These changes in excitability/AP-related parameters likely involve both the Kv7 proteins distributed in dendrites/axons and the AIS, considering the widespread subneuronal distribution patterns of Kv7, although the AIS is the site of the strongest Kv7 enrichment[26,31,33–35]. The suggested notion that Kv7 proteins are important agrees with previous reports indicating that Kv7.2/3 regulates multiple excitability/AP-related parameters that regulate APs[24–27,45–48], and that Kv7.2-mutant mice show increased neuronal excitability and seizure propensity together with behavioral deficits[49,50]. Notably, Kv7.2/3 activation by retigabine rescued the mAHP amplitude but not the other excitability/AP-related parameters. This suggests that retigabine-dependent Kv7 activation can normalize a subset of excitability/AP-related parameters in the context of the mutant mice perhaps after the completion of all compensatory changes, explaining the ~20% (not 100%) survival by the retigabine treatment.

Intriguingly, our results from cultured neurons and expanded brain tissues indicate that the mean intensity of Kv7.3 is decreased in the mutant AIS while the length and area of this region are increased (Fig. 6a–g). Previously, Ank2 has been shown to localize at both sides of the Ank3-enriched AIS and functions as a barrier between the AIS and extra-AIS axonal regions to maintain the normal length of AIS[16,51,52]. In addition, the AIS displays dynamic changes in its position, length, and composition during neuronal development and activity[53,54]. Therefore, Ank2 loss in our mice may weaken the barrier and increase the length of the mutant AIS, thereby decreasing the density and current intensity of Kv7 channels.

Another key result from our study is the overall decrease in Kv7.2/3 levels in the mutant brain (Fig. 5). What might be the underlying molecular mechanisms? Intriguingly, our data indicate that the surface levels of Kv7 channels are decreased in the mutant brain (Fig. 5k). This could be induced by impairments in the trafficking of Kv7 channels such as limited delivery to the surface, suppressed surface stability, enhanced endocytosis, and enhanced lysosomal/degradation pathway. Alternatively, given that Ank2 associates with dynactin (a dynein cargo adapter complex) and PI3-kinase (PIK3C3) to promote fast retrograde axonal transport[17], motor protein-dependent trafficking of Kv7 channels could be disrupted. Altered Kv7 phosphorylation may also play a role, although several phosphorylation sites that we tested did not affect protein stability in cultured neurons. The subcellular sites of decreased Kv7 surface stability could be the AIS, although it could also be non-AIS regions considering that

Kv7 is present at various subcellular sites of neurons, including dendrites, synapses, axons, nodes of Ranvier, and nerve terminals[25,33–36]. Notably, Kv7 proteins are detected in a subset of synapses, and Kv7 levels are moderately decreased at excitatory synapses in mutant neurons (Supplementary Fig. 8a).

Our data indicate that synaptic transmissions are not changed in mutant cortical neurons (Supplementary Fig. 5). However, our proteomics results (PTM, total, and synaptic) indicate that Ank2 proteins are likely present at synaptic sites and regulate synaptic functions (Fig. 5a–i; Supplementary Figs. 6 and 9–12). For instance, Ank2 levels are decreased in total and synaptic proteomes in the mutant brain. In addition, synaptic protein compositions and phosphorylation states are substantially altered in the mutant brain; i.e., upregulated synaptic proteins in the total and synaptic proteomes. Moreover, the mutant synaptic proteome shows a pattern that is opposite to those observed in ASD. Therefore, the spontaneous synaptic transmission that appears to be normal in mutant neurons may be the result of homeostatic/compensatory changes made at the expense of unrecognized synaptic functions.

Ank2 has been implicated in various brain disorders, including ASD and epilepsy[2,3,55,56]. Our Ank2-cKO mice show ASD-related behavioral deficits and epileptiform brain activity with cortical neuronal hyperactivity, resembling human Ank2-related conditions. Our results suggest potassium channel dysfunction (Kv7.2/3) as a novel pathophysiology underlying Ank2-related ASD, providing support for the increasing importance of channelopathies in ASD[57]. However, care should be taken because our mice do not display typical autistic-like phenotypes such as decreased social interaction and increased self-grooming and because retigabine treatment only improves neuronal hyperexcitability, open-field hyperactivity, and seizure-related death, which are associated with the comorbidities but not core symptoms of ASD. Our results also suggest that Kv7 agonists may prove useful as a novel treatment for Ank2-related brain dysfunctions, potentially broadening the usage of Kv7 agonists currently mainly for neurological and mood disorders[58]. Lastly, given that Ank2 has recently been implicated in epilepsy[55,56], our results implicate Kv7 pathophysiology in Ank2-related epilepsy.

In summary, our results together indicate that *Ank2* deletion restricted to cortical excitatory neurons leads to abnormally enhanced neuronal excitability and ASD-related behavioral deficits associated with juvenile seizure-related death through mechanisms, including AIS enlengthening and Kv7.2/3 dysfunction. These findings implicate AIS and Kv7 dysfunctions in the pathophysiology and treatment of Ank2-related brain disorders.

## Methods

### Materials availability
Mouse line generated in this study is available from the lead author on request.

### Animals
C57BL/6J-background transgenic mice harboring a cassette composed of exon 4 of *Ank2* floxed by loxP, along with neomycin-resistance and EGFP genes flanked by Flp-Frt (*Ank2^fl/fl* mice), were generated by

Biocytogen. To generate floxed heterozygous mice ($Ank2^{fl/+}$), the cassette was removed by crossing $Ank2^{fl/fl}$ mice with protamine-Flp mice (C57BL/6 J). To generate Ank2 global knockout (KO) mice, we treated fertilized eggs at the two-cell embryo stage for 30–40 min with purified HTNC, a cell-permeable Cre recombinase (histidine-TAT-nuclear localization-Cre fusion peptide[59]) in the medium at a final concentration of 0.3 mM. To generate Ank2 conditional KO (cKO) mice, $Ank2^{fl/+}$ male mice were crossed with $Emx1$-$Cre$ female mice (Jax005628) to avoid germline transmission of Emx1-Cre. From this mating, we maintained $Emx1$-$Cre;Ank2^{fl/+}$ female mice. Next, $Ank2^{fl/+}$ mice were crossed with each other to produce $Ank2^{fl/fl}$ male mice. Finally, $Emx1$-$Cre;Ank2^{fl/+}$ female mice were crossed with $Ank2^{fl/fl}$ male mice to produce $Ank2^{fl/fl}$ mice (WT) and $Emx1$-$Cre;Ank2^{fl/fl}$ (Ank2-cKO) mice. $Ank2^{fl/fl}$, $Ank2^{fl/+}$, and $Ank2^{+/+}$ mice were genotyped by polymerase chain reaction (PCR) using the following primers: 5'−GCA CAT AGT CAA TGG CGT GA−3' (forward) and 5'−AAA CAC TGC CCA TCA GGA AG−3' (reverse). This primer set amplifies a 267-bp PCR product from the floxed allele and a 215-bp PCR product from the WT allele. $Ank2^{fl/fl}$ and $Emx1$-$Cre;Ank2^{fl/fl}$ mice were genotyped using the following primers: 5'−GAT CTC CGG TAT TGA AAC TCC AGC−3' (forward) and 5'−GCT AAA CAT GCT TCA TCG TCG G−3' (reverse). This primer set produces a 200-bp PCR product in Ank2-cKO mice. Mice were weaned at postnatal day (P) 19 - 22. Up to 8 littermates were co-housed in the same cage. Mice were fed *ad libitum* and housed under a 12-h light/dark cycle. Mouse maintenance was performed according to the Requirements of Animal Research at KAIST. The experimental procedures were approved by the Committee of Animal Research at KAIST (KA2016-32 and KA2020-91). Data from adult male and female $Ank2^{+/-}$ mice were disaggregated in the data presentation because the results were sexually dimorphic. Data from juvenile male and female Ank2-cKO mice were not disaggregated in the data presentation because males and females showed similar behaviors, but the details on male and female mouse numbers were indicated in the Source Data file. Male Ank2-cKO mice were used for acute slice recording, proteomics analysis, and behavioral rescue experiments because males and females showed similar behaviors.

### Behavior tests
Juvenile and adult behavioral tests were performed using both male and female mice. The tests were administered during light-off periods. Subject mice were given at least 1 day of rest between different tests. Data were analyzed either manually or with the EthoVision XT13 program (Noldus).

### Open-field test
A subject mouse was introduced into the center of the open-field box (white acryl chamber, 40 × 40 × 40 cm) and recorded for 1 h (20 min for juvenile open-field test). The center zone was defined as 10 cm apart from the edge (20 × 20 cm). The center zone was illuminated at ~100 lux. Mouse movements were analyzed using EthoVision XT13 (Noldus).

### Elevated plus-maze test
The apparatus for the elevated plus-maze consisted of two closed arms (30 × 5 x 30 cm), two open arms (30 × 5 × 0.5 cm), and a center zone with access to both arms. A whole apparatus was elevated to a height of 50 cm above the floor. Light conditions were 250 lux for open arms and 20 lux for closed arms. A subject mouse was placed in the center zone and allowed to move freely for 8 min. Time spent in open and closed arms was measured using EthoVision XT13 (Noldus)

### Light-dark test
The apparatus for the light-dark test consisted of light (white acryl chamber, 20 cm×30 cm x 20 cm) and dark (black acryl chamber, 20 × 13 × 20 cm) chambers that adhered to each other with an entrance. Light and dark chambers were illuminated at ~600 lux and ~0 lux, respectively. A subject mouse was introduced to the center of the light chamber with their head toward the opposite from the entrance and allowed to freely explore both the light and dark chambers for 10 min. Time spent in the light chamber, activity, and frequency of entrance were measured using EthoVision XT13 (Noldus)

### Repetitive behavior
A subject mouse was placed into a new home cage with bedding and allowed to freely move for 20 min. The center of the cage was illuminated at ~60 lux. Time spent in digging and self-grooming during the last 10 min was measured manually in a blind manner. Digging was defined as a mouse using its head or forelimbs to dig out bedding. Self-grooming was defined as a mouse stroking or scratching its face or body area or licking its body.

### Three-chamber test
Briefly, the subject mice were isolated for 3 days before the test (1 h for the juvenile three-chamber test). Three-chambered apparatus (white acryl chamber, 40 cm × 20 cm × 25 cm) consisted of the left, center, and right chambers arranged in a row, with two entrances to the center chamber[60]. Two containers were located in the corner of the two side chambers. This test consists of three phases, and the duration of each phase was 10 min. In the first phase, the subject was introduced into the center chamber and allowed to freely explore all three chambers with empty containers. In the second phase, the subject mouse was allowed to explore the containers with a stranger1 (S1) or a novel object (O) in the two side chambers. The stranger was randomly positioned in the left or right chamber. Finally, the novel object was replaced with a stranger 2 (S2), and the subject mouse was allowed to explore S1 and S2. During the interval between each session, the subject mouse was gently guided to the center chamber, and the two entrances were blocked. Stranger mice (129SvJae strain for adult subject mice and C57BL/6 J for juvenile subject mice) were sex- and age-matched. The center chamber was illuminated at ~60 lux. Time spent in sniffing the stranger or object was measured using EthoVision XT13 (Noldus).

### Hole-board test
The apparatus (40 cm × 40 cm × 35 cm) consisted of a white acrylic plate with 16 holes (3 cm in diameter, arranged in a 4 × 4 format) and a 50 lux illumination intensity in the center. For 20 min, each mouse was allowed to freely explore the hole-board apparatus. The number of times a mouse poked their nose into a hole was manually counted.

### Ultrasonic vocalization
Male adult subject mice were socially isolated in their home cages for 3 days to allow them to recognize the cage as their own territory. Age-matched C57BL/6 J female adult stranger mice were group-caged for synchronizing the female estrous cycle. A subject male mouse was placed in a novel test cage for 5 min while recording its basal vocalizations in the absence of a female stranger. Next, a stranger female mouse was introduced into the subject cage, and the two mice were allowed to interact freely with each other while recording courtship ultrasonic vocalizations (USVs) of the subject male mouse for 5 min. Avisoft SASLab Pro was used to analyze USVs.

### Morris water maze test
Morris water maze was performed in a round white pool (100 cm diameter, 40 cm deep). The maze was placed in a room with four cues on its walls to indicate directions. Subject mice were trained to find the hidden platform (10 cm diameter) in a maze. Mice were given three trials per day with an inter-trial interval of 30 mi. This test consisted of two phases. First, the learning phase was performed for eight consecutive days, followed by the probe test on day 9 where mice were given 1 min to find the removed platform. Second, for reversal training

(days 10–12), the location of the platform was switched to the opposite position from the previously trained position, and mice were trained to learn the new position of the platform. Mice again performed a probe test on day 13 where mice were given 1 min to find the relocated platform. Target quadrant occupancy during the probe test was measured using EthoVision XT13 (Noldus)

### Contextual fear conditioning test

A day before conditioning day, subject mice were placed in the fear chamber and habituated into the chamber for 5 min. On conditioning day, mice were introduced again to the fear chamber and allowed to freely explore the environment for 2 min, and then received five-foot shocks (0.8 mA, 1-sec interval, 120-sec intervals). After the last shock, the mice were left in the box for additional 2 min, making the total experimental time 12 min. After 24 h, the mice were placed in the same conditioning box and allowed to freely explore for 10 min without any stimuli, and the freezing levels of the mice were quantified. All freezing behaviors were recorded and analyzed using FreezeFrame software (Coulbourn Instruments).

### Rotarod test

The rotarod test was performed using a five-lane rotarod treadmill (Ugo Basile). Mice were trained for 4 consecutive days with three trials per day. One trial lasted for 300 s, while the rotarod was accelerated from 4 to 40 rpm. Mice were carefully placed on each lane of the rotarod and acclimated for 30 s. The latency to fall of each mouse was monitored by an experimenter, and the latency was regarded as 300 s when a mouse withstood the full 300 s.

### Acoustic startle response

Acoustic startle responses were measured using the Med Associates Startle Reflex System (St. Albans). The device consisted of a metal cage atop a series of piezoelectric platforms and a sound system for inducing acoustic startle, both of which were inside a sound-attenuating chamber. Each platform was calibrated using a spinner-type calibrator (Med Associates Startle Calibrator). To test acoustic startle responses, the session was preceded by a 5-min exposure to a 65 dB background noise. Then each mouse received 92 trials with inter-trial intervals ranging from 7 to 23 sec in a pseudorandom order. The trials included a presentation of eight pulse-alone trials (120 dB, 40 ms pulse, four were given at the beginning and four at the end of the test), 77 pulse trials (seven each of 70, 75, 80, 85, 85, 90, 95,100, 105, 110, 115, and 120 dB, 40 ms pulse), and seven trials each without pulse or pre-pulse inhibition. Startle responses at each pulse levels were averaged across trials.

### Pre-pulse inhibition

Pre-pulse inhibition was performed using the same apparatus used for acoustic startle responses. To test pre-pulse inhibition, each mouse received 57 trials with inter-trial intervals ranging from 7 to 23 sec presented in a pseudorandom order. The trials included a presentation of eight pulse-alone trials (120 dB, 40 ms pulse, four were given at the beginning and four at the end of the test), 35 pre-pulse trials (seven each of 70, 75, 80, 85, and 90 dB, 20 ms pre-pulse given 100 ms before a 120 dB, 40 ms pulse), and seven trials each without pulse or pre-pulse presentation. The pre-pulse inhibition percentage was calculated as follows: 100 − (mean pre-pulse response/mean pulse response) × 100.

### Juvenile play

For 1 h, the subject mice (P19–20) were separated from their mother and littermates. Then, in a fresh home cage without bedding, pairs of mice of the same age, sex, and genotype that had never met previously were placed, and their interactions were monitored for 20 min. During the last 10 min, nose-to-nose and nose-to-tail interactions were manually measured in a double-blind manner.

### PTZ-Induced seizure

After intraperitoneal injection of pentylenetetrazole (PTZ; Sigma; 30 mg/kg), subject mice were placed in a clean new home cage. Video recordings for 30 min were used to analyze seizure stages defined as follows; stage 1, behavioral arrest; stage 2, myoclonic (jerk) seizures; stage 3: general tonic-clonic seizures, as previously described[61]. The seizure susceptibility score was defined as follows; 0.2 × 1/(latency to stage 1) + 0.3 × 1/(latency to stage 2) + 0.5 × 1/(latency to stage 3).

### Electroencephalography recording

An electroencephalography (EEG) driver consisted of six stainless steel screws (M1 × 3 mm) and a small header pin connector socket (DF11-6DS-2C, Hirose Electric). All screws were connected with a header pin connector socket through soldered Teflon-coated stainless-steel wire (AS633, Cooner Wire) and implanted on the skull (two for bilateral frontal, +1.8 mm AP, ±1.0 mm ML; two for bilateral parietal, −2.0 mm AP and ±1.8 mm ML from bregma; two for ground and reference, −1.0 mm AP and ±1.0 mm ML from lambda). During the surgery, P20-21 mice were anesthetized with isoflurane (Piramal Healthcare). After a 1-week recovery, EEG recordings were started using a Cheetah Data Acquisition System (Neuralynx) with synchronized video recording. The subject mice were placed into a white acryl box (25 × 25 × 35 cm) with bedding and allowed to freely move around the recording box for 2–3 weeks. In the recording box, mice were fed *ad libitum* and housed under a 12-h light/dark cycle. EEG data were analyzed using a customized MATLAB code.

### Fluorescence in situ hybridization

Frozen mouse brain sections (14 μm thick) were cut coronally through the hippocampal formation, and thaw-mounted onto Superfrost Plus Microscope Slides (Fisher Scientific). The sections were fixed in 4% paraformaldehyde, followed by dehydration in increasing concentrations of ethanol and protease digestion. For hybridization, the sections were incubated in different amplifier solutions in a HybEZ hybridization oven (Advanced Cell Diagnostics) at 40 °C. The probes used in these studies were three synthetic oligonucleotides complementary to the nucleotide sequence 2685–3609 of Mm-Ank2, 62–3113 of Mm-*Gad1*-C2, 552–1506 of Mm-*Gad2*-C3, 464–1415 of Mm-*Slc17a7 (Vglut1)*-C2, and 1986–2998 of Mm-*Slc17a6 (Vglut2)*-C3 (Advanced Cell Diagnostics). The labeled probes were conjugated to Alexa Fluor 488, Atto 550, or Atto 647. The sections were hybridized with probe mixtures at 40 °C for 2 h. Nonspecifically hybridized probes were removed by washing the sections in 1x wash buffer, and the slides were treated with Amplifier 1-FL for 30 min, Amplifier 2-FL for 15 min, Amplifier 3-FL for 30 min, and Amplifier 4 Alt B-FL for 15 min. Each amplifier was removed by washing with 1x wash buffer. The slides were viewed and photographed using TCS SP8 Dichroic/CS (Leica). For the quantification of colocalization, the average number of dots per cell was quantified using the Halo imaging analysis algorithm using the HALO v2.3.2089.18 software (Indica Labs).

### Isotope in situ hybridization

Mouse brain sections (14 μm thick) at embryonic day (E18) and postnatal days (P0, P7, P14, P21, and P56) were prepared using a cryostat (Leica CM 1950). A hybridization probe specific for mouse Ank2 mRNA was prepared using the following regions: nucleotide 315–565 of Ank2 (NM_178655.3). Antisense riboprobe was generated using $^{35}$S-uridine triphosphate (UTP) and the Riboprobe system (Promega).

### Brain slices

After anesthetization with isoflurane (Terrell), mouse brains (P19–22; male) were extracted into ice-cold dissection buffer containing, in mM: 212 D-(+)-sucrose, 25 NaHCO$_3$, 10 D-(+)-glucose, 1.25 L-ascorbic acid, 5 KCl, 1.25 NaH$_2$PO$_4$, 3.5 MgSO$_4$, 2 Na-pyruvate, and 0.5 CaCl$_2$ bubbled with 95% O$_2$ and 5% CO$_2$. Coronal somatosensory slices (300 μm

thickness for whole-cell voltage clamp and 200 μm thickness for multielectrode array) were generated using a vibratome (Leica VT1200) in an ice-cold dissection buffer. The slices were transferred to a 32 °C holding chamber containing artificial cerebrospinal fluid (ACSF) containing in mM (125 NaCl, 25 NaHCO₃, 10 D-(+)-glucose, 2.5 KCl, 1.25 NaH₂PO₄, 2.5 CaCl₂, and 1.3 MgCl₂ bubbled with 95% O₂ and 5% CO₂) and recovered for 30 min. Then, slices were transferred to a recording chamber, where all experiments were performed with circulating ACSF saturated with 95% O₂ and 5% CO₂. After 30-min incubation at room temperature, all experiments using acute brain slices were performed.

## Multielectrode array (MEA)

For multielectrode array recordings, coronal somatosensory slices (200 μm thickness) from P19–22 juvenile male mice were used. After 30-min incubation at room temperature, the slices were gently transferred to a multielectrode probe (MED64 probe, P515A, Panasonic Alpha-Med Sciences), positioning the somatosensory cortex on the electrodes of the MED64 probe. A slice-holding anchor (Slice anchor kit, SHD-22CKIT, Warner Instruments) was used to adhere the slice to the electrode and prevent floating. After loading a brain slice on the MED64 probe, the slice was incubated for 10 min with circulating ACSF at the flow rate of 2 ml/min at 37 °C and 5% CO₂. Then the spontaneous firing was recorded over 10 min. The electrophysiological activity of neurons was analyzed using MATLAB scripts. For the analysis of spike events, recorded signals were filtered with a Gaussian function with a standard deviation of 1.5 ms to suppress high-frequency noises. Spikes were detected with a voltage-threshold-based algorithm; at each electrode, a spike was counted when the filtered signal exceeded a threshold of ±4.5σ, where σ is the standard deviation of pre-processed signal. The refractory period of neurons was assumed to be <1.5 ms and a group of spikes with ISI <10 ms was selected as a burst firing. For the analysis of LFPs, recorded signals were processed using a 150 Hz low-pass filter (Chebyshev filter, 4th order) and LFP events were detected with a voltage-threshold-based algorithm; at each electrode, an LFP event was detected when the fluctuation of filtered signals exceeds a threshold of 4σ where σ is the standard deviation of pre-processed signal. The amplitude of the LFP event was defined as the difference between the minimum and maximum signals, and the duration of LFP was defined as the interval of significant fluctuation ($p < 0.001$, bootstrap) within −200 ms ~ +200 ms of each LFP event. After each recording, the brain slice with the probe on it was subjected to microscopy to capture the position of brain slices and electrodes. The 64 electrodes of the probe were mapped to somatosensory cortex layers using the mouse brain atlas.

## Electrophysiology

MultiClamp 700B amplifier (Molecular Devices) and Digidata 1440 A, 1550 (Molecular Devices) were used for whole-cell voltage-clamp recordings of layer 2/3 pyramidal neurons in the somatosensory cortex. For mEPSC and sEPSC recordings, pipettes (2.5–3.5 MΩ) were filled with an internal solution composed of (in mM: 100 CsMeSO₄, 10 TEA-Cl, 8 NaCl, 10 HEPES, 5 QX-314-Cl, 2 Mg- ATP, 0.3 Na-GTP and 10 EGTA with pH 7.25, 295 mOsm). mEPSC and sEPSC were measured at a holding potential of −70 mV. For mEPSC recordings, ACSF contained tetrodotoxin (0.5 μM) and picrotoxin (60 μM). For sEPSC experiments, 60 μM picrotoxin (Sigma) was additionally added to ACSF. For mIPSC and sIPSC recordings, pipettes (2.5–3.5 MΩ) were filled with an internal solution composed of (in mM: 115 CsCl, 10 TEA-Cl, 8 NaCl, 10 HEPES, 5 Qx- 314-Cl, 4 Mg-ATP, 0.3 Na-GTP, 10 EGTA with pH 7.35, 295 mOsm). For mIPSC recordings, ACSF contained tetrodotoxin (0.5 μM), NBQX (10 μM), and AP5 (50 μM). For sIPSC experiments, NBQX (10 μM), and AP5 (50 μM) were added to ACSF.

To measure the intrinsic excitability of somatosensory cortex layer 2/3 pyramidal neurons, recording pipettes (2.5–3.5 MΩ) were filled with an internal solution composed of (in mM: 137 K-gluconate, 5 KCl, 10 HEPES, 0.2 EGTA, 10 Na-phosphocreatine, 4 Mg-ATP, 0.5 Na-GTP with pH 7.2, 280 mOsm). ACSF contained picrotoxin (60 μM), NBQX (10 μM), and AP5 (50 μM). In this recording, four intrinsic properties were recorded. First, after rupturing and stabilizing cells, currents were clamped, and the resting membrane potential (RMP) was measured. Second, to analyze input resistance, increasing amounts of depolarizing step currents (by 10 pA, −150–10 pA) were injected. The input resistance was calculated as the linear slope of current-voltage plots generated from a series of increasing current injection steps. Next, to evoke single action potential, current inputs were gradually increased from 0 by 10 pA per sweep until inducing single action potential. Finally, minimal currents were injected to hold the membrane potential around −70 mV in a current-clamp mode. To induce a series of action potentials, increasing amounts of depolarizing step currents (by 25 pA, 0–100 pA or by 50 pA, 0–500 pA) were injected.

The amplitude of the medium afterhyperpolarization (mAHP) was defined by a voltage difference between the first 100 ms of the baseline and the minimum point during 500 ms after the end of spikes. dV/dt was calculated as approximate first derivatives by dividing the differences between adjacent voltage values by their time differences. The spike threshold was defined by the voltage at the time point when the second derivative of voltage is at its peak during the voltage rise of the somatic dendritic spike (reference: https://www.ncbi.nlm.nih.gov/pmc/articles/PMC5898699/). The spike amplitude was defined as the voltage difference between the spike peak and threshold. The FWHM was defined as the width of the action potential in the middle of the point between the threshold and the peak of an action potential.

For tonic GABA recording, after rupturing and stabilizing the cell, baseline currents were stabilized with D-AP5 (50 μM) and CNQX (50 μM). The amplitude of the tonic GABA current was measured by the baseline shift after 20 μM bicuculline application. Data were acquired using Clampex 10.2 (Molecular Devices) and analyzed using Clampfit 10 (Molecular Devices). Drugs were purchased from Abcam (TTX), Tocris (Bicuculline, NBQX, CNQX, D-AP5), and Sigma (GABA, picrotoxin).

For bath application of XE991 (Sigma, X2254) or retigabine (Axonmedchem, #1525), the drugs were dissolved in DMSO and added to circulating ACSF for the final concentration of 10 μM. The same volume of DMSO was used as control. To check rescue effects, patched neurons were first recorded in the presence of DMSO followed by 5-min recovery and retigabine treatment.

## PTM analysis

Changes in phosphorylation levels in proteins in the brain of *Emx1-Cre;Ank2ᶠˡ/ᶠˡ* mice were determined using Immobilized Metal Affinity Chromatography (IMAC) Service (Cell Signaling Technology). Briefly, the cortex and hippocampus of a mouse were extracted in an ice-cold buffer containing protease/phosphatase inhibitors. Brain samples from three different mice were pooled together to make n number one. Brain samples were snap-frozen in liquid nitrogen, protease-digested, and fractionated by solid-phase extraction. The fractionated peptides were incubated with designated immobilized PTM (post-translational modification)-motif antibodies, and the peptides containing the corresponding PTM sequences were eluted and analyzed using LC-MS/MS. Mass spectra were assigned to peptide sequences using the Sorcerer program. Finally, the peptide sequence assignment was linked to parent ion peak intensities to measure approximate fold changes in validated peptides between paired samples.

## Total and synaptosomal proteomic analysis

Total (whole lysates) and crude synaptosomal samples from the cortex and hippocampus of WT and Ank2-cKO mice (3 weeks) were

lysed with 1x sodium dodecyl sulfate (SDS) buffer (5% SDS, 50 mM triethylammonium bicarbonate (TEAB), pH 8.5). The lysates were purified and digested using the S-trap method following the provided protocol. Tryptic-digested peptides were labeled with 16-plex TMT isotopes (Thermo Fisher Scientific). After labeling, all TMT-labeled peptides were combined and dried in Speed-Vac and desalted using Pierce peptide desalting spin columns (Thermo Fisher Scientific). Then peptides were combined and fractionated into 20 fractions using basic reverse-phase liquid chromatography. LC-MS/MS analysis was performed using an UltiMate 3000 RSLCnano system (Thermo Scientific) coupled to Orbitrap Exploris 480 mass spectrometer (Thermo Fisher Scientific). Mobile phases A and B were composed of 0 and 99.9% acetonitrile containing 0.1% formic acid, respectively. The LC gradient at a flow rate of 250 nl/min was applied for 140 min for peptide separation. The Orbitrap Exploris 480 was operated in data-dependent mode, and MS2 scans were performed with HCD fragmentation (37.5% collision energy). MS/MS spectra were identified and quantified using Integrated Proteomics Pipeline software and the UniProt mouse database. Search parameters were precursor mass tolerance of 5 ppm, a fragment ion mass tolerance of 600 ppm, two and more peptide assignments for protein identification at a false positive rate of <0.01, and TMT reporter ion mass tolerance of 20 ppm. Statistics and data analysis were conducted using Perseus software (version 1.6.15). The expressions of proteins between samples were compared with Welch's $t$-test with a $p$-value set at <0.05.

## PSEA analysis

Gene Set Enrichment Analysis (GSEA) (http://software.broadinstitute.org/gsea) using DEPs (PSEA) was applied using the list of all proteins expressed, ranked by the fold change, and multiplied by the inverse of the P value. Enrichment analysis was performed using GSEAPreranked (gsea-4.2.3.jar) module on gene set collections downloaded from Molecular Signature Database (MSigDB) v7.5.1 (http://software.broadinstitute.org/gsea/msigdb). The gene sets with an FDR of <0.05 were considered significantly enriched.

## Western blot

Mice were anesthetized with isoflurane and decapitated for brain extraction. The cortex and hippocampus were isolated on ice and acutely homogenized with ice-cold brain homogenization buffer containing 0.32 M sucrose, 10 mM HEPES, pH 7.4, 2 mM EDTA, 2 mM EGTA, protease inhibitors, and phosphatase inhibitors. Total lysates were prepared by boiling the homogenates with β-mercaptoethanol.

## Immunoprecipitation

WT mouse cultured hippocampal neurons at DIV (DIV) 17–18 ($1 \times 10^6$ cells/60 mm dish) were rinsed with ice-cold Tyrode's solution (136 mM NaCl, 2.5 mM KCl, 2 mM CaCl2, 1.3 mM MgCl2, 10 mM Na-HEPES, 10 mM D-glucose, pH 7.3) before lysis. Washed neurons were lysed with 600 μl lysis buffer (150 mM NaCl, 50 mM Tris pH7.4, 1 mM EDTA, 1% TX-100, protease and phosphatase inhibitor cocktail (Thermo, 78445, 1:100)) and clarified by centrifugation at 20,000 × g for 10 min at 4 °C followed by sonication for 5 seconds, and then transfer the supernatants to new microcentrifuge tube without disturbing the pellet. The supernatants were incubated with 1 μg of primary antibodies (mouse IgG, anti-Ank2 antibody, StressMarq, AB_11232606; anti-Ank3 antibody, Santa Cruz Biotechnology, sc-12719) at 4 °C for 2 h under rotator. The antibody-antigen complexes were then incubated with 20 μl protein A/G agarose beads (Santa Cruz Biotechnology, SC2003) for 1 h at 4 °C under gentle rotation. Beads were washed three times in lysis buffer to remove non-specific binding. The immunoprecipitated proteins were eluted from beads by heating for 30 min at

50 °C in SDS-PAGE sample loading buffer. The eluted proteins were then resolved by SDS-PAGE and immunoblotted.

## Surface biotinylation

Cultured neurons from WT and Ank2-cKO mice at DIV 18 were biotinylated using EZ-Link Sulfo-NHS-LC-Biotin (Pierce; 1 mg/ml) for 1 h at 4 °C. After washing with Tyrode's solution three times, neurons were incubated with 100 mM glycine/Tyrode's solution for 30 min at 4 °C to quench unbound Sulfo-NHS-SS-biotin. Then, cultured neurons were washed in Tyrode's solution and lysed in a lysis buffer (Tris-Cl pH 7.4 40 mM, NaCl 120 mM, EDTA 1 mM, 1% Triton X-100, and protease inhibitor) and sonicated briefly. Then homogenates were centrifuged at 18,000 × g at 4 °C for 20 min, and the supernatant was incubated with NeutrAvidin Agarose Resin (Thermo). The precipitated surface proteins were eluted by heating for 30 min at 50 °C.

## Immunocytochemistry

Cultured mouse neurons at DIV 14–28 were fixed in 4% paraformaldehyde/4% sucrose/Tyrode's solution, permeabilized in 0.25% Triton X-100/Tyrode's solution for 5 min, and incubated in 10% normal donkey serum/Tyrode's solution. Cells were then incubated with primary antibodies in 3% NDS/Tyrode's solution for 2 h at 37 °C: mouse Ank3 (Sant Cruz Biotechnology, sc-12719, 1:200), rabbit Kv7.3 (Alomone, APC-051, 1:1000), chicken mCherry (Abcam, ab205402, 1:5000), mouse PSD-95 (NeuroMab, 75-028, 1:1000), and mouse gephyrin (Synaptic Systems, 147 111, 1:1000). Then, the following secondary antibodies were diluted in 3% NDS/Tyrode's solution and used: donkey anti-mouse IgG Alexa Fluor 488 (Thermo, A21202, 1:1000), donkey anti-rabbit IgG Alexa Fluor 405 (Thermo, A48258, 1:1000), goat anti-chicken IgY Alexa Fluor 594 (Thermo, A11042, 1:1000), and donkey anti-rabbit IgG Alexa Fluor 594 (Thermo, A21207, 1:1000). Immunofluorescence images were acquired using a confocal microscope (63× objective; LSM780; Carl Zeiss). The images were analyzed using ImageJ (Fiji).

## Brain tissue expansion and immunohistochemistry

Seven WT and seven cKO mouse brains were extracted after perfusion. Ice-cold heparin in PBS with 4% paraformaldehyde was administered transcardially to remove the blood and fix the brain. The extracted brains were further fixed in 4% PFA solution at 4 °C, overnight. The samples were then washed with PBS at 4 °C for 12 h followed by additional washing for 12 h at room temperature. The brains were incubated in eMAP solution for 1 week at 4 °C[62]. Then, the brain immersed in eMAP solution underwent gelation at 35 °C, overnight. The gel-embedded brains were then sectioned at 150 μm with a vibrating microtome. Both WT and cKO eMAP-processed 150-μm-thick tissues were incubated with either rabbit anti-Ank3 antibody (1:300 dilution) or rabbit anti-Caspr antibody (1:300 dilution) with 2% normal donkey serum in PBS and 0.1% (wt/vol) Triton X-100 (PBST) at 37 °C for 12 h, followed by PBST washing three times for 1 h each. The tissues were then incubated with Alexa Fluor 647-conjugated anti-rabbit IgG antibody (1:300 dilution) and DAPI (1:10,000 dilution) in PBST at 37 °C for 12 h, followed by PBST washing three times for 1 h each. The samples were immersed in deionized water for expansion. The expanded samples were mounted in between a slide glass and a coverslip using a spacer filled with deionized water. Samples were imaged with a confocal microscope (Olympus FV3000). A 60×, 1.2-NA water-immersion objective (UPLSAPO60XW) was used to image neurons at layers 2/3 of the somatosensory cortex. The images were visualized and analyzed with Fiji and Imaris 9.7.2 (Bitplane). To measure the length and surface area of the AIS, individual neurons were cropped in Imaris with the assistance of the 3D crop tool in Fiji. The AIS of each neuron was segmented through the semi-automatic surface segmentation algorithm in Imaris. The length of the AIS was determined by measuring the

distance between two points or multi-points according to the axon's curvature. Fifteen neurons were measured for each of four WT and four cKO mouse brain samples. The length of the node of Ranvier was measured manually using Imaris by calculating the distance between the two center points of the outer boundary of the paranodes.

### RT-qPCR

To determine mRNA levels, RNAs were extracted from the whole cortex of WT and Ank2-cKO mice at P7, P14, P21, and P28 using TRIzol™ Reagent (Ambion, 15596026). cDNAs were synthesized using M-MLV cDNA synthesis kit (Enzynomics, EZ006M), and real-time qPCR (RT-qPCR) was performed using THUNDERBIRD SYBR qPCR Mix (Toyobo, QPS-201) and CFX96 Real-Time System. For RT-qPCR, the primers in the Key Resources Table were used. The Gapdh gene was used as a control gene.

### Drug treatment

For chronic retigabine treatment, WT and Ank2-cKO mice were provided *ad libitum* access to drinking water containing retigabine (Axonmedchem, #1525) starting at early weaning at P16. The final concentration of retigabine treatment was 20 mg/kg per day. Retigabine-containing water was replaced every 2 days. During the course of drug administration, WT and Ank2-cKO mice were subjected to the open-field test at P21. For a shortened chronic treatment, the water containing retigabine was replaced with normal water at P42. For acute retigabine treatment, WT and Ank2-cKO mice were injected intraperitoneally with retigabine dihydrochloride (Axonmedchem, #2252) in phosphate-buffered saline (5 mg/kg), or phosphate-buffered saline alone (control). The mice were subjected to behavioral tests after a 10-min rest period in their home cages after injection. Mice were divided into two groups; retigabine or saline first.

### Quantification, statistical analysis, and reproducibility

Statistical analyses were performed using the GraphPad Prism 7 software (RRID:SCR_002798). Briefly, the normality of the data distribution was determined using the D'Agostino and Pearson normality test ($n = 8$ or greater) or the Shapiro-Wilk normality test ($n < 8$). When the data showed a normal distribution, probability values were calculated using the Student's $t$-test. When the data did not show a normal distribution, the Mann–Whitney $U$-test was used for probability values. Additionally, if the data did not show equal variance, Welch's correction was used. For two-way ANOVA (repeated-measures), if an interaction had a meaningful p-value, Sidak's multiple comparisons test or Tukey's multiple comparisons test was used for post-hoc comparisons. Data are presented as means ± standard error of the mean (SEM). $N$ numbers are indicated in each figure legend. Statistical significances are indicated at the end of each figure, as follows: *$p < 0.05$, **$p < 0.01$, and ***$p < 0.001$. Statistical details are presented in Supplementary Dataset 1. All experiments were replicated through multiple cohort/mice analysis, where applicable. We included the results only when the replications lead to the same conclusions.

### Key resources table

| Reagent or resource | Source | Identifier |
|---|---|---|
| **Antibodies (dilution)** | | |
| Mouse anti-Ank2 (1:500) | Stress Marq | Cat.# SMC-400, RRID: AB_11232606 |
| Mouse anti-Ank3 (1:1000 for WB, 1:100 for ICC) | Santa Cruz Biotech | Cat.# sc-12719, RRID: AB_626674 |
| Rabbit anti-Ank3 (1:300) | Synaptic Systems | Cat.# 386 003, RRID: AB_2661876 |
| Rabbit anti-Kv1.2 (1:1000) | Millipore | Cat.# Ab5924, RRID: AB_92137 |
| Mouse anti-Kv2.1 (1:500) | NeuroMab | Cat.# 73-014, RRID: AB_10672653 |
| Rabbit anti-Kv3.1b (1:500) | Alomone | Cat.# APC-014, RRID: AB_2040166 |
| Rabbit anti-Kv7.2 (1:1000) | Abcam | Cat.# ab22897, RRID: AB_775890 |
| Rabbit anti-Kv7.3 (1:500) | Alomone | Cat.# APC-051, RRID: AB_2040103 |
| Rabbit anti-GABA$_A$R γ2 (1:1000) | Synaptic Systems | Cat.# 224 003, RRID: AB_2263066 |
| Mouse anti-GABA$_A$R β2/3 (1:500) | Millipore | Cat.# MAB341, RRID: AB_2109419 |
| Mouse anti-Gephyrin (1:1000) | Synaptic Systems | Cat.# 147 111, RRID: AB_887719 |
| Mouse anti-PSD95 (1:1000) | NeuroMab | Cat.# 75-028, RRID: AB_2292909 |
| Rabbit anti-Nav1.2 (1:200) | Alomone | Cat.# ASC-002, RRID: AB_2040005 |
| Rabbit anti-Nav1.6 (1:1000) | Alomone | Cat.# ASC-009, RRID: AB_2040202 |
| Rabbit anti-Caspr (1:300) | Abcam | Cat.# ab34151, RRID: AB_869934 |
| Anti-mouse GFP (1:1000) | Santa Cruz Biotech | Cat.# sc-9996, RRID: AB_627695 |
| Anti-mouse Myc (1:1000) | Cell Signaling | Cat.# 2276, RRID: AB_331783 |
| Chicken anti-mCherry (1:5000) | Abcam | Cat.# ab205402, RRID: AB_2722769 |
| Rabbit anti-GAPDH (1:1000) | Cell Signaling | Cat.# 2118, RRID: AB_561053 |
| Mouse anti-GAPDH (1:1000) | Cell Signaling | Cat.# 97166, RRID: AB_275682 |
| Mouse anti-α-tubulin (1:1000) | Sigma | Cat.# T5168, RRID: AB_477579 |
| Mouse anti-β-actin (1:1000) | Sigma | Cat.# 5316, RRID: AB_476743 |
| Donkey anti-mouse IgG Alexa Fluor 488 (1:1000) | Thermo | Cat.# A21202, RRID: AB_141607 |
| Donkey anti-rabbit IgG Alexa Fluor 405 (1:1000) | Thermo | Cat.# A48258, RRID: AB_2890547 |
| Goat anti-chicken IgY Alexa Fluor 594 (1:1000) | Thermo | Cat.# A11042 RRID: AB_2534099 |
| Donkey anti-rabbit IgG Alexa Fluor 594 (1:1000) | Thermo | Cat.# A21207, RRID: AB_141637 |
| Donkey anti-rabbit IgG Alexa Fluor Plus 647 (1:300) | Thermo | Cat.# A32795, RRID: AB_2762835 |
| **Bacterial and virus strains** | | |
| Cre-P2A-mCherry (AAV serotype PHP.eB) | This paper | |
| mCherry (AAV serotype PHP.eB) | This paper | |
| **Chemicals, peptides, and recombinant proteins** | | |
| Picrotoxin | Abcam | Cat.# ab120315, CAS: 124-87-8 |
| NBQX | Tocris | Cat.# 1044/50, CAS: 479347-86-9 |
| CNQX | Tocris | Cat.# 1045, CAS: 479347-85-8 |
| D-AP5 | Tocris | Cat.# 0106, CAS: 79055-68-8 |
| Bicuculline | Tocris | Cat.# 0109, CAS: 66016-70-4 |
| γ-Aminobutyric acid | Sigma | Cat.# A2129 CAS: 56-12-2 |
| Tetrodotoxin | Abcam | Cat.# ab120055, CAS: 18660-81-6 |

| XE-991 | Sigma | Cat.# X2254, CAS: 122955-42-4 |
|---|---|---|
| Retigabine | Axonmedchem | Cat.# 1525 CAS: 150812-12-7 |
| Retigabine dihydrochloride | Axonmedchem | Cat.# 2252 CAS: 150812-13-8 |
| **Experimental models: organisms/strains** | | |
| *Emx1-Cre* mice | Jackson Labs | JAX_005628,RRID:IMSR_JAX:005628 |
| Ank2$^{floxed/+}$ mice | This paper | N/A |
| **Oligonucleotides** | | |
| Ank2 floxed Forward primer 5'-GCA CAT AGT CAA TGG CGT GA-3' | This paper | N/A |
| Ank2 floxed Reverse primer 5'-AAA CAC TGC CCA TCA GGA AG-3' | This paper | N/A |
| Emx1-Cre Forward primer 5'-GAT CTC CGG TAT TGA AAC TCC AGC-3' | This paper | N/A |
| Emx1-Cre Reverse primer 5'-GCT AAA CAT GCT TCA TCG TCG G-3' | This paper | N/A |
| Ank2 Probe (2685-3609 of Mm-Ank2) | Advanced Cell Diagnostics (ACD) | Cat #: 413221 |
| GAD1 Probe (62-3113 of Mm-GAD1-C3) | Advanced Cell Diagnostics (ACD) | Cat #: 400951-C2 |
| GAD2 Probe (552-1506 of Mm-GAD2-C2) | Advanced Cell Diagnostics (ACD) | Cat #: 415071-C3 |
| Vglut1 Probe (464-1415 of Mm-Slc17a7-C2) | Advanced Cell Diagnostics (ACD) | Cat #: 416631-C2 |
| Vglut2 Probe (1986–2998 of Mm-Slc17a6-C3) | Advanced Cell Diagnostics (ACD) | Cat #: 319171-C3 |
| Ank2 qRT-PCR Forward primer 5'-ATC GGA GTC AGA TCA AGA GCC G-3' | This paper | N/A |
| Ank2 qRT-PCR Reverse primer 5'-AAG CCA GCC TTT CTT CCA TCC G-3' | This paper | N/A |
| Kcnq2 qRT-PCR Forward primer 5'-AGT CCA AGA GCA GCA TCG GCA A-3' | This paper | N/A |
| Kcnq2 qRT-PCR Reverse primer 5'-CAG TGA CTG TCC GCT CGT AGT A-3' | This paper | N/A |
| Kcnq3 qRT-PCR Forward primer 5'-AAG CCT ACG CTT TCT GGC AGA G-3' | This paper | N/A |
| Kcnq3 qRT-PCR Reverse primer 5'-ACA GCT CGG ATG GCA GCC TTT A-3' | This paper | N/A |
| Gapdh qRT-PCR Forward primer 5'-CAT CAC TGC CAC CCA GAA GAC TG-3' | This paper | N/A |
| Gapdh qRT-PCR Reverse primer 5'-ATG CCA GTG AGC TTC CCG TTC AG-3' | This paper | N/A |

| **Software and algorithms** | | |
|---|---|---|
| GraphPad Prism 7 | GraphPad | RRID: SCR_002798 |
| EthoVision XT 13.0 software | Noldus | RRID: SCR_000441 |
| Avisoft SASLab Pro software | Avisoft | RRID: SCR_014438 |
| Image J | National Institutes of Health (NIH) | RRID: SCR_003070 |
| MATLAB | MathWorks | RRID: SCR_001622 |
| Imaris 9.7.2 | Bitplane | RRID: SCR_007370 |
| Cytoscape | Cytoscape | RRID: SCR_003032 |
| HALO 2.3 | HALO | RRID: SCR_018350 |
| BioRender | BioRender | RRID: SCR_018361 |

## Reporting summary

Further information on research design is available in the Nature Portfolio Reporting Summary linked to this article.

## Data availability

Source Data are provided with this paper. All proteomics data generated in this study were deposited in the ProteomeXchange Consortium via the MassIVE partner repository with the accession code PXD040968 (total and synaptosomal proteome) and via the PRIDE partner repository with the accession code PXD041769 (PTM proteome). Source data are provided with this paper.

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

## Acknowledgements
This work was supported by the National Research Foundation (NRF) grant (2020M3E5D9080794 to H.Kim; 2022R1A2C3008991 to S.P.; 2021R1A2C2005294 to T.K.; 2022M3H9A2096186 to J.Y.K.), the Korea Institute of Science and Technology Information (K-21-L02-C10 to H.Kang), and the Institute for Basic Science, Korea (IBS-R002-D1 to E.K. and IBS-R001-D2 to C.J.L.).

## Author contributions
H.O., Y.O., S.K., and R.K. performed behavioral experiments; H.O., Y.O., Y.E.Y., H.C., W.K., and W.W. performed electrophysiological experiments; H.O. and Y.O. performed immunoblot experiments; H.O. and Se.L. performed EEG experiments; Su.L. performed IP and ICC experiments; H.O. and W.C. performed MEA experiments; E.Y. performed isotope in situ hybridization and FISH experiments; Y.S.K. performed brain tissue expansion and immunohistochemistry experiments; Y.Y. performed proteomics experiments; H.K. performed PSEA analysis. H.O., C.J.L., H.Ki., H.Ka., S.-B.P., J.Y.K., T.K., and E.K. designed the experiments and wrote the manuscript.

## Competing interests
The authors declare no competing interests.
