## [Peer Review File · Nature Communications]

Kv7/KCNQ potassium channels in cortical hyperexcitability and juvenile seizure-related death in Ank2-mutant miceREVIEWER COMMENTS

Reviewer #1 (Remarks to the Author):

Oh and colleagues investigate the most frequently mutated ASD-risk gene *Ank2*. They do this by generating a new *Ank2* conditional knockout using the *Emx2-Cre*. They perform an impressive and comprehensive set of behavioral and electrophysiological analyses. These are all outstanding. They show ASD-like behaviors and altered neuronal excitability. They also perform a comprehensive proteomic analysis of post-translational modifications. Among these, they focused on the K⁺ channels due to their over-representation in their proteomic data set and the altered physiology. Among these, immunoblotting analyses showed Kv7.2 and Kv7.3 as being the most changed. Since these channels are found at AIS, they show a slightly longer AIS in the cKO and reduced mean channel intensity. Therefore, they considered whether Kv7 agonists could rescue some of the phenotypes observed. They found that among the phenotypes, the seizures and some of the premature death could be rescued through the use of retigabine. However, none of the ASD-like behavioral abnormalities was rescued. Overall, the data are solid and compelling. I have only two concerns:

1. There is no clear explanation for why Kv7 channels are altered in the *Ank2* cKO mouse. Kv7 channels interact with AnkB at the AIS. The authors speculate the longer AIS may reflect the disruption of the distal AIS boundary. However, that doesn't completely explain the relatively large decrease in total Kv7 channel protein. Kv7 channels have a similar ankyrin binding motif as that found in Na⁺ channels so it may not be surprising that AnkB and Kv7 channels interact ... just not at the AIS. Kv7 channels may also be found in mossy fibers and other brain regions. It is possible the altered excitability has nothing to do with AIS properties. Is Kv7 channel localization disrupted in other brain regions where AnkB is also found? Does this channel colocalize with AnkB in any other brain region?
2. Throughout the discussion the authors make some statements that I think are over-interpretation. For example, the authors state "Our results suggest potassium channel dysfunction(Kv7.2/3) as a novel pathophysiology underlying *Ank2*-related ASD." It is important to emphasize the authors have only shown the link between altered Kv7.2/3 and the epilepsy-related phenotypes. They further suggest "...broadening the usage of Kv7 agonists from neurological and mood disorders to neurodevelopmental disorders such as ASD." I think that these statements are unwarranted and I encourage the authors to soften the statements about KV7 channels in any of the behavioral aspects of ASD.

Reviewer #2 (Remarks to the Author):

Manuscript: Nature Com

"Kv7/KCNQ potassium channels in cortical hyperexcitability and juvenile seizure-related death in *Ank2*-mutant mice" by Oh et al..

The paper by Oh et al describes the generation and characterization of several ANK2 mouse models that reflect human disease related mutations.

This is an important and comprehensive study that thoroughly analysis the impact of an ANK2 deletion (especially in cortex and hippocampus) on the neuronal excitability, molecular composition of neuronal compartments as well as behaviour. The study is based on a wide set of methods that create a very good view on different sites of action of the ANK2 protein. The study provides new data on molecular mechanisms as well as treatment options for ANK2 related neuropsychiatric disorders including their comorbidities and is therefore of very high translational value. Therefore, it is very significant for the field.

Major:

-How does the newly designed mouse KO/mutation relate to genetic alterations in ASD /ANK2 patients/cases and their clinical picture/phenotype? What models have been designed so far and how do they differ from each other? This should be included in the Introduction and/or Discussion

part.

-In line with this: the mouse model introduced here, does not present all core symptoms of autism, like reduced social interaction and enhanced self-grooming, however important co-morbidities. This should be mentioned/discussed and the authors should be more careful in saying that the potassium channel dysfunction is a novel explanation for the ASD phenotype since only the hyperexcitability and survival could be rescued by the correction of the "channelopathy".

-There is to me a bit of an experimental gap between data obtained up to Fig 5 and then Fig 6. The authors detect especially alterations from genes known to be within the synaptic compartment by phosphoproteomic analysis and confirmed these changes by western using homogenate and synaptosomal fractions. Then they concentrated (certainly for good reasons) on the AIS. What about alterations of Kv channels (and ANK2) at (inhibitory/excitatory) synaptic sites? Data on this would be an important bridge leading to the exciting data on AIS alterations. (What about nodes of Ranvier?)

-By the extensive phosphoproteomic analysis the authors found an important co-regulation of certain Kv channels and ANK2. However, it is still not clear how this co-regulation can be explained. It would be very nice to further employ the experimental in vitro set-up from Fig 6 and analyse the temporal changes of Kv7.3 expression (structural alterations) according to the time point of ANK2 depletion and ... mRNA levels over time would be very helpful for further interpretation.

-I understand that the Kv7.3 antibody does not work after the expansion procedure. However, it would be important to show the reduction of Kv7.3 in the in vivo mouse model as well. Can this be done by regular fixation and immunostaining without expansion (as indicated by the neuronal cell culture date)?

- Could the authors comment on the number of regulated genes (phosphoproteomics) that are known to be ASD related? and in line ...Have the authors confirmed i.e. Shank2 downregulation that might explain the hyperactivity of the mice that cannot be corrected by Retigabine?

Minor:

-what is the effect of Retigabine on the increased social behaviour and lowered repetitive behaviour?

-Fig 6a Stainings are hard to see and judge at least in my print outs (can the pictures be enlarged?)

-The authors should add more animals to the immune/expansion experiments (N2 is indicated in the MAT/MET section)

-the authors should explain why residual ANK2 bands can be detected in the KO animals.

-Figures are not labelled with FIG1-.....

Fig. 1/Suppl Fig 2 spelling mistake: "Open field" ..

Fig 5 Spelling mistake: "Cortactin"

I support the publication of this study after revision according the points raised here.

Reviewer #3 (Remarks to the Author):

Oh et al 2022 characterize a new conditional knock out mouse targeting Ank2. This mouse eliminates both 220 and 440kDa ankyrin-B upon Cre-expression based on removal of exon 4 of ank2. They first conducted a battery of behavioral analysis of global Ank2+/- mice to find that they were largely normal. However, Emx-1-Cre driven homozygous deletion of Ank2 showed autism like phenotypes. These mice also have spontaneous seizures causing death in young adulthood. They delve into the mechanism behind this seizure related death and show that there is increased firing by multi-electrode array recordings and increased excitability by patch-clamp

recordings. They attribute this increased neuronal excitability to decreased Kv7.2/3 which they identify in an unbiased proteomic PTM screen. In cultured neurons, the AIS morphology is elongated and exhibit increased neuronal excitability. They utilize a FDA approved Kv7 agonist to rescue the seizure related death. While this treatment did normalize the electrophysiological properties measured, it only caused a 20% survival in the mice and no improvement on behavioral measures. In conclusion, this study utilized multiple levels of analysis from protein biochemistry, electrophysiology, cell biology, through animal behavior to investigate a novel mechanism of neuronal excitability regulation through ank2 regulation of Kv7.2/3. These results are significant and improve our understanding of disease mechanisms related to ank2.

EMX-1Cre is not specific for excitatory neurons.

The authors do not address the rationale for generating a new ank2 floxed mouse as previous mice have targeted similar shared exons of ank2 such as exon 22. It is undiscussed why behavioral results are different between the exon22 and exon4 heterozygous mice. Of note, the studies reported here are done during light-off periods and only on adult males.

Proteomic PTM based analysis revealed large numbers of proteins with significant changes. It would be of interest if this analysis was done without a focus on PTM modified proteins as the levels of potassium channels themselves have decreased expression based on their Western blot analysis.

The authors postulate that novel Ser 410/438 and Ser 457 on Kv 7.2 and 7.3 respectively could mediate protein stability. This should be explicitly tested. The authors also introduce the idea that Kv7.2/3 protein stability is not based on direct interaction with ank2. Does this function through Kv

While the morphology of the AIS is changed in the cultured ank2 null neurons, the protein levels of both ankyrin-G and Kv7.3 are unchanged. The width of the AIS appears widened in the representative images.

In the RTG rescue experiment, it appears that the mice that live to ~25 days seem to continue to live. This potentially suggests a critical window where this mechanism is necessary between early postnatal life and PND25.

Western blots of ankyrins show low levels of high molecular weight isoforms. The authors do not list the types of gels utilized and could be the reason why they appear this way.

The AIS measurements were done with n=15 per mouse. These measurements are highly variable and the n should be significantly higher in order to make these claims.

Figure 1a is not drawn to scale.

Authors write that the mice showed altered anxiety-like behaviors in OF, L/D, and EPM. However, they exhibit both increased and decreased anxiety behaviors in the different assays and this is not discussed.

Quantification of Supplementary Fig 1c and d would help to understand levels of colocalization.

Re: NCOMMS-22-18932-T

Point-by-point responses to reviewers' comments

Reviewer #1 (Remarks to the Author):

Oh and colleagues investigate the most frequently mutated ASD-risk gene Ank2. They do this by generating a new Ank2 conditional knockout using the Emx2-Cre. They perform an impressive and comprehensive set of behavioral and electrophysiological analyses. These are all outstanding. They show ASD-like behaviors and altered neuronal excitability. They also perform a comprehensive proteomic analysis of post-translational modifications. Among these, they focused on the K⁺ channels due to their over-representation in their proteomic data set and the altered physiology. Among these, immunoblotting analyses showed Kv7.2 and Kv7.3 as being the most changed. Since these channels are found at AIS, they show a slightly longer AIS in the cKO and reduced mean channel intensity. Therefore, they considered whether Kv7 agonists could rescue some of the phenotypes observed. They found that among the phenotypes, the seizures and some of the premature death could be rescued through the use of retigabine. However, none of the ASD-like behavioral abnormalities was rescued. Overall, the data are solid and compelling. I have only two concerns:

Ans: We sincerely appreciate the positive summary and thoughtful comments of the reviewer.

1. There is no clear explanation for why Kv7 channels are altered in the Ank2 cKO mouse. Kv7 channels interact with AnkG at the AIS. The authors speculate the longer AIS may reflect the disruption of the distal AIS boundary. However, that doesn't completely explain the relatively large decrease in total Kv7 channel protein. Kv7 channels have a similar ankyrin binding motif as that found in Na⁺ channels so it may not be surprising that AnkB and Kv7 channels interact ... just not at the AIS. Kv7 channels may also be found in mossy fibers and other brain regions. It is possible the altered excitability has nothing to do with AIS properties. Is Kv7 channel localization disrupted in other brain regions where AnkB is also found? Does this channel colocalize with AnkB in any other brain region?

Ans: We appreciate these thoughtful comments. It is indeed important to better understand how Kv7 protein levels are decreased. We agree with the reviewer that the increased AIS length and decreased density of Kv7 channels in the AIS may not fully explain the strong decrease in Kv7 levels.

Kv7 proteins could be closely associated with Ank2 proteins. However, we could not obtain such evidence thus far, as supported by the lack of coimmunoprecipitation of the two proteins in cultured neurons (**Supplementary Fig. 7c**). In addition, we could not detect coimmunoprecipitation of Ank2 with Ank3 that directly interacts with Kv7 (**Supplementary Fig. 7d**).

We agree that Kv7 and Ank2 interacting in non-AIS neuronal compartments may be important. We found during revision that Kv7 channels are present at small fractions of synaptic sites, and the levels of Kv7 channels at excitatory synapses are

moderately decreased (**Supplementary Fig. 8a**), suggesting that a synaptic decrease of Kv7 may contribute to the neuronal hyperactivity. However, when we performed a proteomic analysis of total (whole-lysates) and crude synaptosomal samples from Ank2-cKO mice, Kv7 proteins were more strongly decreased in the total proteome relative to the synaptosomal proteome (**Fig. 5i and Supplementary Fig. 9c**), suggesting that the synaptic decrease of Kv7 may be less important.

Kv7 could interact with Ank2 in other brain regions given the widespread expression patterns of Kv7.2/3 and Ank2 in the brain. We have to point out, however, that the Emx1-Cre driver in Ank2-cKO mice mainly deletes Ank2 in cortical and hippocampal regions, and that we used cortical + hippocampal samples to demonstrate the decreases in Kv7 proteins (**Fig. 5j**).

To further gain insights into how Kv7 proteins are decreased, we determined temporal courses of the changes in Ank2 and Kv7.2/3 mRNA/protein levels at multiple postnatal stages. Ank2 and Kv7.2/3 proteins were strongly decreased at around postnatal day/P 14 in the WT brain, and both Ank2 and Kv7.2/3 began to decrease at around P14 without changes in the mRNA levels (**Supplementary Fig. 13**). Therefore, the decrease in Ank2 proteins (not mRNAs) is temporally associated with the decrease in Kv7.2/3 proteins.

Importantly, we found that the surface levels of Kv7.2 proteins are decreased in the mutant brain, as supported by the surface biotinylation results (**Fig. 5k**), suggesting that Ank2 promotes surface expression of Kv7.

These results collectively suggest that Ank2 does not closely associate with Kv7 but indirectly regulates the surface expression of Kv7 to maintain protein stability. In addition, synaptic populations of Kv7 seem to be less important for the decreased Kv7 levels. We added these results and discussion to the revised manuscript.

2. Throughout the discussion the authors make some statements that I think are over-interpretation. For example, the authors state “Our results suggest potassium channel dysfunction(Kv7.2/3) as a novel pathophysiology underlying Ank2-related ASD.” It is important to emphasize the authors have only shown the link between altered Kv7.2/3 and the epilepsy-related phenotypes. They further suggest “...broadening the usage of Kv7 agonists from neurological and mood disorders to neurodevelopmental disorders such as ASD.” I think that these statements are unwarranted and I encourage the authors to soften the statements about KV7 channels in any of the behavioral aspects of ASD.

Ans: We agree with the reviewer and toned down related texts in the Discussion.

Reviewer #2 (Remarks to the Author):

Manuscript: Nature Com

"Kv7/KCNQ potassium channels in cortical hyperexcitability and juvenile seizure-related death in Ank2-mutant mice" by Oh et al..

The paper by Oh et al describes the generation and characterization of several ANK2 mouse models that reflect human disease related mutations.

This is an important and comprehensive study that thoroughly analysis the impact of an ANK2 deletion (especially in cortex and hippocampus) on the neuronal excitability, molecular composition of neuronal compartments as well as behaviour. The study is based on a wide set of methods that create a very good view on different sites of action of the ANK2 protein. The study provides new data on molecular mechanisms as well as treatment options for ANK2 related neuropsychiatric disorders including their comorbidities and is therefore of very high translational value. Therefore, it is very significant for the field.

Ans: We sincerely appreciate the positive summary and thoughtful comments of the reviewer.

Major:

1. How does the newly designed mouse KO/mutation relate to genetic alterations in ASD/ANK2 patients/cases and their clinical picture/phenotype? What models have been designed so far and how do they differ from each other? This should be included in the Introduction and/or Discussion part.

Ans: We appreciate this thoughtful comment. We attempted to delete exon 4 of the *Ank2* gene to delete both the short and long forms of the Ank2 protein (220 and 440 kDa with shared N-terminus; see **Fig. 1a**) and thus induce stronger phenotypes. This was clarified in the revised Results.

An early study on Ank2-mutant mice that lack both the short and long Ank2 variants displayed neonatal lethality (Scotland *et al.*, 1998), making it difficult to assess behavioral changes. A more recent study characterized three Ank2-mutant lines; the first two lack only the 440-kDa band while the third (lacking exon 22) lacks both the short and long variants with the latter being similar to our mice in design (also lacking both short and long variants via exon 4 deletion) (Yang *et al.*, 2019).

Exon 4 and 22 heterozygous *Ank2* deletions lead to distinct phenotypes. Our male *Ank2*^{+/-} mice (exon 4; adult) show largely normal behaviors, except for modest anxiety-like behavior (decreased open-field center time) (**Supplementary Fig. 2**). Our revision experiments on female Ank2 mice (exon 4; adult) also indicated that behaviors are largely normal except for decreased self-grooming (**Supplementary Fig. 3**). In the previous study (Yang *et al.*, 2019), Ank2 mice (exon 22; male) show impaired urine marking in adult males and suppressed ultrasonic vocalizations in juvenile males (the comparisons summarized in **Supplementary Table 7**). Therefore, exon 4 and 22 Ank2-mutant mice seem to show distinctly altered behaviors in the tests that do not overlap between the two studies, which should be clarified in additional studies. These aspects were detailed in the revised Discussion.

2. In line with this: the mouse model introduced here, does not present all core symptoms of autism, like reduced social interaction and enhanced self-grooming, however important co-morbidities. This should be mentioned/discussed and the authors should be more careful in saying that the potassium channel dysfunction is a novel explanation for the ASD phenotype since only the hyperexcitability and survival

could be rescued by the correction of the “channelopathy”.

Ans: we fully agree with the reviewer and modified the texts in the Discussion to clearly state that our mice do not display typical ASD-related social and repetitive behavioral deficits and that only neuronal hyperexcitability, hyperactivity (open-field; obtained during revision), and seizure-related death were rescued. In addition, we toned down our claim that channelopathy represents a new pathophysiology for core symptoms of ASD.

3. There is to me a bit of an experimental gap between data obtained up to Fig 5 and then Fig 6. The authors detect especially alterations from genes known to be within the synaptic compartment by phosphoproteomic analysis and confirmed these changes by western using homogenate and synaptosomal fractions. Then they concentrated (certainly for good reasons) on the AIS. What about alterations of Kv channels (and ANK2) at (inhibitory/excitatory) synaptic sites? Data on this would be an important bridge leading to the exciting data on AIS alterations. (What about nodes of Ranvier?)

Ans: In response, we tested if synaptically localized Kv7 channels are altered in Ank2-mutant cultured neurons by immunostaining. We found that Kv7 is localized at small fractions (~5–10%) of excitatory and inhibitory synapses in cultured neurons and that Kv7 levels are moderately decreased at excitatory but not at inhibitory synapses in cultured Ank2-cKO neurons (**Supplementary Fig. 8a**), suggesting that Ank2 deletion decreases synaptic levels of Kv7.

However, when we performed proteomic analyses using total (whole-lysates) and crude synaptosomal samples (in addition to the phosphoproteomic/PTM analysis in the original manuscript), the decrease in Kv7 levels was stronger in total lysates relative to crude synaptosomes (**Fig. 5i; Supplementary Fig. 9c**), suggesting that synaptic Kv7 is less important for the hyperexcitability of mutant neurons.

Whether Ank2 proteins are present at synaptic sites could not be determined because of the lack of a suitable antibody, but Ank2 proteins were decreased at similar levels in total and crude synaptosomal proteomes from Ank2-cKO mice (**Fig. 5h; Supplementary Fig. 9b**), suggesting that Ank2 could be present at synapses and regulate synaptic functions.

Lastly, we found by additional experiments that the length of the node of Ranvier, where Kv7 proteins are also present, was unchanged in expanded Ank2-cKO brain slices (**Supplementary Fig. 8b**), dissimilar to the enlengthened AIS in the mutant brain.

4. By the extensive phosphoproteomic analysis the authors found an important co-regulation of certain Kv channels and ANK2. However, it is still not clear how this co-regulation can be explained. It would be very nice to further employ the experimental in vitro set-up from Fig 6 and analyse the temporal changes of Kv7.3 expression (structural alterations) according to the time point of ANK2 depletion and ... mRNA levels over time would be very helpful for further interpretation.

Ans: In response, we determined the temporal courses of the changes in Ank2 and

Kv7.2/3 mRNA/protein levels at multiple postnatal stages. We used in vivo brain slices because Ank2 and Kv7.2/3 protein levels begin to be strongly increased at ~P14 in WT mouse brains (**Supplementary Fig. 13a-c**), suggesting that the maximal effects of Ank2 deletion are likely to occur starting from ~P14.

In the mutant brain, Kv7.2/3 mRNA levels were stable across P7, P14, P21, and P28, but Kv7.2/3 proteins began to be strongly decreased starting from ~P14, a time point when Ank2 also began to be strongly decreased (**Supplementary Fig. 13d-i**). These results suggest that Kv7.2/3 mRNA levels are not an important factor, but the decrease in Ank2 proteins is temporally correlated with and thus may induce the decrease in Kv7.2/3 proteins.

Importantly, we found during revision a decrease in the surface levels of Kv7 proteins in the mutant brain, as supported by biotinylation experiments using cultured neurons (**Fig. 5k**), suggesting that Ank2 deletion may decrease Kv7 levels via the destabilization of Kv7 at the membrane surface.

5. I understand that the Kv7.3 antibody does not work after the expansion procedure. However, it would be important to show the reduction of Kv7.3 in the in vivo mouse model as well. Can this be done by regular fixation and immunostaining without expansion (as indicated by the neuronal cell culture date)?

Ans: We tried to label Kv7.3 using un-expanded mouse brain slices. Unfortunately, we were unable to obtain reliable signals under these conditions.

6. Could the authors comment on the number of regulated genes (phosphoproteomics) that are known to be ASD related? and in line ...Have the authors confirmed i.e. Shank2 downregulation that might explain the hyperactivity of the mice that cannot be corrected by Retigabine?

Ans: In response, we analyzed the overlaps between the PTM, total, and synaptosomal DEPs and ASD-risk genes (**Supplementary Fig. 12d-h**). The numbers of overlapped ASD-risk genes were particularly high in downregulated total DEPs (~2.1-folds relative to upregulated total DEPs), upregulated synaptosomal DEPs (~2.8-folds relative to downregulated synaptosomal DEPs), and downregulated PTM proteins (~7.4-folds relative to upregulated PTM proteins). Many of the overlapped proteins were also SynGO proteins.

We appreciate your interesting idea that Shank2 may be involved. Shank2 levels, however, were modestly but significantly increased (not decreased) in total DEPs ($p = 0.0415$; fold change = 1.056). In addition, while the hyperactivity was not rescued by chronic retigabine treatment, an acute retigabine treatment attempted during revision did rescue hyperactivity (**Fig. 8h-k**), where we did not measure drug-induced changes in Shank2 levels for the acute nature of the treatment.

Minor:

1. what is the effect of Retigabine on the increased social behaviour and lowered repetitive behaviour?

Ans: We performed related experiments and found that retigabine has no effects on

repetitive behaviors (self-grooming and digging) or juvenile play (**Supplementary Fig. 15b–f**).

2. Fig 6a Stainings are hard to see and judge at least in my print outs (can the pictures be enlarged?)

Ans: We made the images clearer.

3. The authors should add more animals to the immune/expansion experiments (N2 is indicated in the MAT/MET section)

Ans: We doubled mouse and neuron numbers to 4 mice and 60 neurons and reached the same conclusion (**Fig. 6f,g**).

4. the authors should explain why residual ANK2 bands can be detected in the KO animals.

Ans: It is probably because Ank2 is expressed in inhibitory neurons in addition to excitatory neurons (**Supplementary Fig. 1d-f**); Emx1 drives gene expression in excitatory neurons and glial cells but not GABAergic inhibitory neurons (Gorski *et al*, 2002). We commented on this in Results.

5. Figures are not labelled with FIG1-.....

Ans: We added figure numbers in the revised figures.

6. Fig. 1/Suppl Fig 2 spelling mistake: “Open field” ..

Ans: Corrected; We appreciate it.

7. Fig 5 Spelling mistake: “Cortactin”

Ans: Corrected; We appreciate it.

8. I support the publication of this study after revision according the points raised here.

Ans: We appreciate again for the very helpful comments.

Reviewer #3 (Remarks to the Author):

Oh et al 2022 characterize a new conditional knock out mouse targeting Ank2. This mouse eliminates both 220 and 440kDa ankyrin-B upon Cre-expression based on removal of exon 4 of ank2. They first conducted a battery of behavioral analysis of global Ank2^{+/-} mice to find that they were largely normal. However, Emx-1-Cre driven homozygous deletion of Ank2 showed autism like phenotypes. These mice also have spontaneous seizures causing death in young adulthood. They delve into

the mechanism behind this seizure related death and show that there is increased firing by multi-electrode array recordings and increased excitability by patch-clamp recordings. They attribute this increased neuronal excitability to decreased Kv7.2/3 which they identify in an unbiased proteomic PTM screen. In cultured neurons, the AIS morphology is elongated and exhibit increased neuronal excitability. They utilize a FDA approved Kv7 agonist to rescue the seizure related death. While this treatment did normalize the electrophysiological properties measured, it only caused a 20% survival in the mice and no improvement on behavioral measures. In conclusion, this study utilized multiple levels of analysis from protein biochemistry, electrophysiology, cell biology, through animal behavior to investigate a novel mechanism of neuronal excitability regulation through ank2 regulation of Kv7.2/3. These results are significant and improve our understanding of disease mechanisms related to ank2.

Ans: We sincerely appreciate the positive summary and thoughtful comments of the reviewer.

1. EMX-1Cre is not specific for excitatory neurons.

Ans: The reviewer is correct. Emx1-Cre drives gene expression in glial cells in addition to excitatory neurons although not in GABAergic neurons. We clarified this with a reference in the Results.

2. The authors do not address the rationale for generating a new ank2 floxed mouse as previous mice have targeted similar shared exons of ank2 such as exon 22. It is undiscussed why behavioral results are different between the exon22 and exon4 heterozygous mice. Of note, the studies reported here are done during light-off periods and only on adult males.

Ans: We generated a new *Ank2*-floxed mouse line to try both global and conditional Ank2 KO, which is now clarified in Results. Both exon 4 and exon 22 deletions indeed lead to the loss of both the 220- and 440-kDa variants of the Ank2 protein. However, different patient mutations located in different regions of the *ANK2* gene are expected to give different phenotypes because different mutations in the same gene frequently induce different phenotypes in mice (i.e., Shank3). We clarified this rationale in the revised Results.

In addition, the current and previous results indicate that *Ank2* exon 4 and 22 heterozygous deletions lead to distinct phenotypes. Our *Ank2*^{+/-} mice (exon 4; adult male) show largely normal behaviors, except for modest anxiety-like behavior (decreased open-field center time) (**Supplementary Fig. 2**). Our experiments on female *Ank2*^{+/-} mice (adult), performed during revision, also indicate largely normal behaviors except for decreased self-grooming (**Supplementary Fig. 3**). In the previous study (Yang *et al.*, 2019), Ank2 mice (exon 22; male) show impaired urine marking in adult males and suppressed ultrasonic vocalizations in juvenile males (the comparisons summarized in **Supplementary Table 7**). Therefore, Ank2 exon 4- and 22-mutant mice seem to show distinctly altered behaviors in the tests that do not overlap between the two studies, making it less useful to compare the distinct

phenotypes. However, if the same set of behavioral tests were performed for the two mouse lines (exon 4 and 22) and the behaviors turn out to be different, potential reasons for the difference could be light-on/off state or age (juvenile vs. adult), as correctly pointed out by the reviewer, or the nature deleted exons (exon 4 vs. 22). With regard to two other mouse lines tested in the previous study (Yang *et al.*, 2019), these mice lack only the 440- but not 220-kDa band, making it not meaningful to compare the phenotypes with ours. These aspects were detailed in the revised Discussion.

3. Proteomic PTM based analysis revealed large numbers of proteins with significant changes. It would be of interest if this analysis was done without a focus on PTM modified proteins as the levels of potassium channels themselves have decreased expression based on their Western blot analysis.

Ans: The levels of Kv7.2/3 PTMs in Ank2-cKO mice from the proteomic analysis range ~21–63% of WT levels (25-63% for Kv7.2 and 21% for Kv7.3) (**Fig. 5f**) whereas those for Kv7.2/3 proteins determined by immunoblotting ranges ~50–75% of WT levels (**Fig. 5j**). In addition, levels of Kv7.2/3 determined by the proteomic analysis of total (whole-lysate) proteins (not PTM) performed during revision also ranges ~81–87% of WT levels (**Fig. 5i**). These results suggest that both the decreases in total and PTM levels contribute to the final changes in PTM levels. We clarified this in the revised Results.

4. The authors postulate that novel Ser 410/438 and Ser 457 on Kv 7.2 and 7.3 respectively could mediate protein stability. This should be explicitly tested. The authors also introduce the idea that Kv7.2/3 protein stability is not based on direct interaction with ank2. Does this function through Kv... (the review comment stopped here, but we tried to address this point; see our responses below.)

Ans: We appreciate these comments. In response, we generated Kv7.2 and Kv7.3 mutants mimicking phosphorylation and dephosphorylation (Kv7.2-S410-E/A, Kv7.2-S438-E/A, and Kv7.3-S457-E/A) and found that these mutations do not affect protein stability in cultured neurons (**Supplementary Fig. 7b**), excluding the possibility that they may regulate protein stability.

In addition, we tested if the surface expression levels of Kv7.2/3 are changed by biotinylation experiments and found decreases in the surface expression of Kv7.2 (**Fig. 5k**). This suggests that Ank2 regulates the surface stability of Kv7 proteins, which is unlikely to occur through a direct interaction of Kv7 with Ank2 (**Supplementary Fig. 7c**).

5. While the morphology of the AIS is changed in the cultured ank2 null neurons, the protein levels of both ankyrin-G and Kv7.3 are unchanged. The width of the AIS appears widened in the representative images.

Ans: In response, we analyzed the width of the AIS and found that it was not altered (**Fig. 6b**).

6. In the RTG rescue experiment, it appears that the mice that live to ~25 days seem

to continue to live. This potentially suggests a critical window where this mechanism is necessary between early postnatal life and PND25.

Ans: The apparent ~PND25 (postnatal day 25) is actually ~PND40; we apologize for the confusing labeling of the x-axis; it was days after drug administration, but not PND, which started from ~PND16. However, the reviewer's idea on the potential presence of a critical period was intriguing. We thus stopped the drug treatment rather early at ~PND42 (~26 days after the initiation of the treatment), narrowing down the treatment period to PND16–42. Intriguingly, the mutant mice that survived up until PND42 all died after the cessation of the drug treatment (**Fig. 8c,d**), suggesting that the retigabine treatment does not have long-lasting effects and thus that a critical period of the treatment may not exist.

7. Western blots of ankyrins show low levels of high molecular weight isoforms. The authors do not list the types of gels utilized and could be the reason why they appear this way.

Ans: We replaced the 440 kDa band image with a better one (**Supplementary Fig. 1b**), which we obtained using a gradient PAGE gel; we appreciate this comment.

8. The AIS measurements were done with n=15 per mouse. These measurements are highly variable and the n should be significantly higher in order to make these claims.

Ans: We doubled mouse and neuron numbers to 4 mice and 60 neurons and reached the same conclusion (**Fig. 6f,g**).

9. Figure 1a is not drawn to scale.

Ans: We redrew **Fig. 1a** to scale.

10. Authors write that the mice showed altered anxiety-like behaviors in OF, L/D, and EPM. However, they exhibit both increased and decreased anxiety behaviors in the different assays and this is not discussed.

Ans: In response, we generated correlograms comparing the distance moved and anxiety measures (center/light-chamber/open-arm time) in the three tests (open-field, light-dark, and elevated plus-maze). The distance moved in the mutant mice did not correlate with the measures of anxiety-like behaviors in the three tests (**Supplementary Fig. 4**), suggesting that the hyperactivity in the mutant mice is unlikely to function as a confounding factor in the observed anxiety/anxiolytic-like behaviors. Ank2-cKO mice seem to show different anxiety-like behaviors in different tests likely because of the different nature of the anxiogenic stimuli (i.e., open area, height, and light); this was commented in Results.

11. Quantification of Supplementary Fig 1c and d would help to understand levels of colocalization.

Ans: We quantified the results in **Supplementary Fig. 1d** and found that nearly all glutamatergic and GABAergic neurons in the cortex and hippocampus were Ank2-positive. In the converse quantifications, large portions of Ank2-positive neurons are glutamatergic neurons rather than GABAergic neurons (**Supplementary Fig. 1e,f**), reflecting that glutamate neurons are more abundant than GABAergic neurons. **Supplementary Fig. 1c** shows isotope in situ data only for Ank2 but not glutamatergic/GABAergic markers, making it infeasible to assess colocalization.

References

- Gorski JA, Talley T, Qiu M, Puelles L, Rubenstein JL, Jones KR (2002) Cortical excitatory neurons and glia, but not GABAergic neurons, are produced in the Emx1-expressing lineage. *J Neurosci* 22: 6309-6314
- Scotland P, Zhou D, Benveniste H, Bennett V (1998) Nervous system defects of AnkyrinB (-/-) mice suggest functional overlap between the cell adhesion molecule L1 and 440-kD AnkyrinB in premyelinated axons. *J Cell Biol* 143: 1305-1315
- Yang R, Walder-Christensen KK, Kim N, Wu D, Lorenzo DN, Badea A, Jiang YH, Yin HH, Wetsel WC, Bennett V (2019) ANK2 autism mutation targeting giant ankyrin-B promotes axon branching and ectopic connectivity. *Proc Natl Acad Sci U S A* 116: 15262-15271

REVIEWER COMMENTS

Reviewer #1 (Remarks to the Author):

The authors have performed an admirable number of additional experiments and made extensive efforts to address the concerns of all reviewers. I was impressed with the study before, and am even more so now. There are still unanswered questions, but the authors have attempted to address many of them. Future studies, beyond the current work, may ultimately provide answers. I strongly support publication of this paper.

Reviewer #2 (Remarks to the Author):

The authors have answered all my questions and concerns convincingly and I fully support publication ...
Tobias Boeckers

Reviewer #3 (Remarks to the Author):

Oh et al has satisfactorily responded to all comments from the first review.

Supplemental 1b. 440kDa Ank2 western blot raw image shows significant protein degradation but less decrease in the Het sample in comparison to the cKO from 1b. In addition, 3 out of 5 WT samples on the blot have no detectable 440kDa band. Considering the WT and HET bands selected have less B-actin signal, it is unclear why this would be the case. I would consider reacquisition of the sample with high denaturing conditions and rerunning this gel. Additionally, there seems to be differences in migration of the. 440kDa band in comparison to the ladder. However, I do not think it affects the interpretation as the authors are assuming the protein is indeed present in the WT.

A discussion as to the meaning of decreased repetitive behaviors should be added. Grooming and digging behavior in the WT mice is not considered pathological and are naturalistic behaviors important for health and food acquisition. Is it that the cKO mice are constantly moving such that they cannot perform grooming and digging behavior. Is the lack of grooming behavior affecting the fur of the mice?

Mice with a truncated 440kDa but intact 220kDa Ankyrin-B had no detectable seizure activity as measured by multisite LFP recordings (see Mague et al 2022). This would strongly implicate the 220kDa isoform as being the major contributor to the striking findings reported in figure 2.

The decrease in membrane surface levels of Kv7 could be through destabilization of Kv7 at the membrane or decreased trafficking. As ank2 is already implicated in trafficking, I think it is plausible that this could be happening as well. (see Lorenzo et al 2014).

Re: NCOMMS-22-18932A

Point-by-point responses to reviewers' comments

Reviewer #1 (Remarks to the Author):

The authors have performed an admirable number of additional experiments and made extensive efforts to address the concerns of all reviewers. I was impressed with the study before, and am even more so now. There are still unanswered questions, but the authors have attempted to address many of them. Future studies, beyond the current work, may ultimately provide answers. I strongly support publication of this paper.

Ans: We sincerely appreciate your positive assessment of the revised manuscript.

Reviewer #2 (Remarks to the Author):

The authors have answered all my questions and concerns convincingly and I fully support publication ...
Tobias Boeckers

Ans: We sincerely appreciate your positive assessment of the revised manuscript.

Reviewer #3 (Remarks to the Author):

Oh et al has satisfactorily responded to all comments from the first review.

Ans: We sincerely appreciate your positive assessment of the revised manuscript.

1. Supplemental 1b. 440kDa Ank2 western blot raw image shows significant protein degradation but less decrease in the Het sample in comparison to the cKO from 1b. In addition, 3 out of 5 WT samples on the blot have no detectable 440kDa band. Considering the WT and HET bands selected have less B-actin signal, it is unclear why this would be the case. I would consider reacquisition of the sample with high denaturing conditions and rerunning this gel. Additionally, there seems to be differences in migration of the 440kDa band in comparison to the ladder. However, I do not think it affects the interpretation as the authors are assuming the protein is indeed present in the WT.

Ans: We appreciate these careful comments. In response, we optimized immunoblot experimental conditions for the results shown in **Supplementary Fig. 1b**. Now both 440 and 220 kDa bands are clearly seen in all five samples (**Supplementary Fig. 1b**; see also the raw images in the **Source Data** file). In addition, the 440 kDa bands are now tight around the size marker (~460 kDa) (**Source Data**).

The fact that Ank2 protein levels in Ank2-cKO mice are at ~10% of WT levels (**Fig. 1b**) suggest a stronger expression of *Ank2* in Emx1-positive excitatory neurons

relative to other cell types and a stronger impact of their loss by Cre-dependent homozygous (not heterozygous) gene deletion.

2. A discussion as to the meaning of decreased repetitive behaviors should be added. Grooming and digging behavior in the WT mice is not considered pathological and are naturalistic behaviors important for health and food acquisition. Is it that the cKO mice are constantly moving such that they cannot perform grooming and digging behavior. Is the lack of grooming behavior affecting the fur of the mice?

Ans: In response, we tested the suggested correlations and found that the decreased repetitive behaviors (self-grooming and digging) did not correlate with the hyperactivity of the mutant mice (**Supplementary Fig. 4f,g**), suggesting that hyperactivity did not confound the repeat-behavioral results.

With regard to the fur of mice, we did not notice a detectable genotype difference.

3. Mice with a truncated 440kDa but intact 220kDa Ankyrin-B had no detectable seizure activity as measured by multisite LFP recordings (see Mague et al 2022). This would strongly implicate the 220kDa isoform as being the major contributor to the striking findings reported in figure 2.

Ans: We appreciate this comment. We have to point out, however, that the Ank2-HT mice used in the Mague et al. paper (2022) differs from our Ank2-cKO mice, which represent a homozygous Ank2-mutant mice and again display much stronger reductions of both 440- and 220-kDa Ank2 variants (~10% of WT levels). A more comparable mouse line in our study is Ank2-HT mice that lacks both 440- and 220-kDa Ank2 variants. However, these mice display no seizure activity and normal behaviors both in males and females (**Supplementary Figs 2 and 3**), similar to the absence of seizure activity in the Ank2-HT mice used in the Mague et al. paper. Therefore, the loss of the 220-kDa band is unlikely to underlie the seizure phenotype of Ank2-cKO mice. Rather, the strong decrease in both 440- and 220-kDa Ank2 variants (~10% of WT levels) may be responsible for the seizure phenotype. We commented on these aspects in the revised discussion as follows: "In addition, a more recent study on *Ank2*^{+/-} mice lacking only the 440 kDa Ank2 variant (P2580fs line)¹² did not show seizure activity⁴³, similar to our *Ank2*^{+/-} mice (**Supplementary Figs. 2 and 3**). Therefore, the stronger decline of Ank2 proteins (~10% of WT levels for both 440- and 220-kDa variants) rather than the type of Ank2 variants lost seem to cause the seizure phenotype in Ank2-mutant mice."

4. The decrease in membrane surface levels of Kv7 could be through destabilization of Kv7 at the membrane or decreased trafficking. As ank2 is already implicated in trafficking, I think it is plausible that this could be happening as well. (see Lorenzo et al 2014).

Ans: We appreciate telling us about this possibility. We added the following sentence to the revised discussion: "Alternatively, given that Ank2 associates with dynactin (a dynein cargo adaptor complex) and PI3-kinase (PIK3C3) to promote fast retrograde

axonal transport ¹⁷, motor protein-dependent trafficking of Kv7 channels could be disrupted.”